# Golden Syrian hamster as a model to study cardiovascular complications associated with SARS-CoV-2 infection

Zaigham Abbas Rizvi[1,2]*†, Rajdeep Dalal[1]†, Srikanth Sadhu[1]†, Akshay Binayke[1], Jyotsna Dandotiya[1], Yashwant Kumar[3], Tripti Shrivastava[4], Sonu Kumar Gupta[3], Suruchi Aggarwal[3], Manas Ranjan Tripathy[1], Deepak Kumar Rathore[4], Amit Kumar Yadav[3], Guruprasad R Medigeshi[4], Amit Kumar Pandey[4], Sweety Samal[4], Shailendra Asthana[3], Amit Awasthi[1,2]*

[1]Immuno-biology Lab, Infection and Immunology Centre, Translational Health Science and Technology Institute, NCR-Biotech Science Cluster, Faridabad, India; [2]Immunology Core, Translational Health Science and Technology Institute, NCR-Biotech Science Cluster, Faridabad, India; [3]Non-communicable Disease Centre, Translational Health Science and Technology Institute, NCR-Biotech Science Cluster, Faridabad, India; [4]Infection and Immunology Centre, Translational Health Science and Technology Institute, NCR-Biotech Science Cluster, Faridabad, India

*For correspondence:
zaigham.abbas15@gmail.com
(ZAR);
aawasthi@thsti.res.in (AA)

†These authors contributed
equally to this work

Competing interest: The authors
declare that no competing
interests exist.

Reviewing Editor: Shiv Pillai,
Ragon Institute, United States

**Abstract** Severe acute respiratory syndrome coronavirus-2 (SARS-CoV-2) infection in the Golden Syrian hamster causes lung pathology that resembles human coronavirus disease (COVID-19). However, extrapulmonary pathologies associated with SARS-CoV-2 infection and post-COVID sequelae remain to be understood. Here, we show, using a hamster model, that the early phase of SARS-CoV-2 infection leads to an acute inflammatory response and lung pathologies, while the late phase of infection causes cardiovascular complications (CVCs) characterized by ventricular wall thickening associated with increased ventricular mass/body mass ratio and interstitial coronary fibrosis. Molecular profiling further substantiated our findings of CVC as SARS-CoV-2-infected hamsters showed elevated levels of serum cardiac troponin I, cholesterol, low-density lipoprotein, and long-chain fatty acid triglycerides. Serum metabolomics profiling of SARS-CoV-2-infected hamsters identified N-acetylneuraminate, a functional metabolite found to be associated with CVC, as a metabolic marker found to be common between SARS-CoV-2-infected hamsters and COVID-19 patients. Together, we propose hamsters as a suitable animal model to study post-COVID sequelae associated with CVC, which could be extended to therapeutic interventions.

## Editor's evaluation

This study is of broad interest since it does provide data on cardiovascular disease in a hamster model of SARS-CoV-2 disease. It also makes interesting novel connections to lipidomic and metabolomic profiles in the context of the complications of a virally induced disease.

## Introduction

First reported in Wuhan, China, in December 2019, severe acute respiratory syndrome coronavirus-2 (SARS-CoV-2) has infected nearly 0.9% of the total world population with a mortality rate of around 1.96% as of December 2021 (https://COVID-19.who.int/). Symptomatic COVID-19 is typically characterized by symptoms ranging from mild to acute respiratory distress associated with cytokine release

syndrome (*Chen and Li, 2020*; *Verity et al., 2020*). Moreover, extrapulmonary signs, ranging from cardiovascular complication (CVC), coagulopathies, multiple-organ damage, and neurological disorders, were found to be recognized as post-COVID sequelae described in patients with mild to severe COVID-19 (*Chen and Li, 2020*; *Cortinovis et al., 2021*; *Francis et al., 2021*; *Huang et al., 2021*; *Lamers et al., 2020*; *Mao et al., 2020*; *Neurath, 2020*; *Nishiga et al., 2020*; *Wu et al., 2020b*; *Xiao et al., 2020*; *Xydakis et al., 2020*). The severity of COVID-19 is governed by several factors such as titer of virus, route of viral entry, gender, age, and comorbid conditions (*Chen and Li, 2020*; *Sungnak et al., 2020*; *Wang et al., 2020*). The receptor-binding domain (RBD) of the spike (S) glycoprotein of SARS-CoV-2 engages with angiotensin-converting enzyme 2 (ACE2), a cellular receptor expressed primarily on host lung epithelial cells, which facilitates viral entry into the host cell. In addition, other factors such as cellular transmembrane protease 'serine 2' (TMPRSS2) and neuropilin-1 (NRP-1) assist in the cellular entry of SARS-CoV-2 (*Cantuti-Castelvetri et al., 2020*; *Daly et al., 2020*).

Various nonhuman primates and small animal models have been described to study the pathogenesis and transmission of SARS-CoV-2 infection (*Callaway, 2020*; *Cohen, 2020*; *Johansen et al., 2020*; *Lakdawala and Menachery, 2020*; *Shi et al., 2020*). The Golden Syrian hamster, which was previously described as a model for SARS-CoV infection, has gained much attention as a suitable model for studying SARS-CoV-2 infection (*Chan et al., 2020*; *Imai et al., 2020*; *Kaptein et al., 2020*; *Kreye et al., 2020*; *Lee et al., 2020*; *Osterrieder et al., 2020*; *Rosenke et al., 2020*; *Sia et al., 2020*; *Tostanoski et al., 2020*). Remarkably, hamsters have been shown to be infected through intranasal, oral, and ophthalmic routes by SARS-CoV-2 with respectively descending lung viral load and pathologies (*Imai et al., 2020*; *Lee et al., 2020*). While several reports have described the usefulness of the hamster model in preclinical evaluations of vaccines and therapeutics for COVID-19 as well as studying SARS-CoV-2-associated lungs pathology, only limited studies have focused on extrapulmonary pathologies that are associated with post-COVID sequelae in hamsters.

Here, we studied SARS-CoV-2-associated pulmonary and extra pathologies in the Golden Syrian hamster. Following intranasal infection, hamsters showed a high lung viral load with significantly increased lungs injuries on days 2 and 4 post infection (dpi). Our data show an acute immune reaction characterized by heightened expression of inflammatory cytokines in the early phase of infection. Strikingly, at the later phase of the infection, the cardiomyocytes of the infected hamsters showed a high viral load with interstitial coronary fibrosis. SARS-CoV-2 infection increased the thickening of ventricular walls and the interventricular septum of the heart, with an increased serum troponin I level. Cardiovascular changes, upon 4 dpi, were found to be associated with an increase in serum triglycerides (TGs), low-density lipoprotein (LDL), cholesterol, and high-density lipoprotein (HDL) while long-chain fatty acids (LCFAs) were found to be increased on 7 dpi. Furthermore, our metabolomics data identified changes in serum metabolites of the infected, as compared to uninfected, hamsters. In the line with data on CVC, we found an increased serum N-acetylneuraminate, a functional metabolite found to be associated with coronary artery disease (CAD) and myocardial injury (*Maciel et al., 2016*; *Zhang et al., 2018*), in the samples of the SARS-CoV-2-infected hamsters and COVID-19 patients. These data further substantiated our claim that SARS-CoV-2 infection increases serum lipids and metabolites that are associated with CVC. Taken together, here we provide shreds of evidence that the SARS-CoV-2 infection in hamsters results in post-COVID sequelae marked by CVC with elevated levels of molecular markers of CVC. We thus propose the Golden Syrian hamster as an appropriate and valuable model to study both early SARS-CoV-2 infection and postinfection sequelae.

## Results

### SARS-CoV-2 infection in hamsters results in acute inflammatory response and lung pathology

Previous studies have shown that the interactions of SARS-CoV-2 RBD and a fragment of S1 protein with host ACE2 receptor and NRP-1 were critical for virus entry into epithelial cells (*Chan et al., 2020*; *Hoffmann et al., 2020*). Since SARS-CoV-2 infection in hamsters mimics human infection, therefore it was essential to understand the similarities between human (hu) and hamster (ha) receptors, especially ACE2 and NRP-1, and study their interactions with SARS-CoV-2 at the molecular level. To compare amino acid residues of hu and ha ACE2 that interact with the RBD, structure-guided sequence alignment was performed with their respective sequences (*Figure 1—figure supplement 1A*). The overlay

of the crystal structure of huACE2 with modeled haACE2 was 1.2 Å, indicating the high structural resemblance (*Figure 1—figure supplement 1B and C*). At the interface, 20 residues of hu ACE2 interact with 17 residues of RBD while 20 residues of ha ACE2 interact with 16 residues of RBD (*Figure 1—figure supplement 1D and E*). The NRP-1 receptor, which binds to S1 fragment of the spike protein and facilitates SARS-CoV-2 entry into cells (*Cantuti-Castelvetri et al., 2020*; *Daly et al., 2020*), also showed identical interaction with S1 fragment residues between hu and ha (*Figure 1— figure supplement 1F–I*).

The in silico data provided a comparative molecular insight into SARS-CoV-2 interaction and cellular entry in hamsters, which showed molecular and structural similarities of SARS-CoV-2 interaction and cellular entry with humans. Previous reports on SARS-CoV-2 infection in hamsters have shown a decrease in body mass, upon SARS-CoV-2 infection, as one of the clinical signs for a successful establishment of infection. In line with this, our data showed that hamsters challenged with either $10^4$ or $10^5$ plaque-forming unit (PFU) SARS-CoV-2 showed a gradual decrease in body weight up to 4 dpi, and subsequently recovered. The body weight loss peaked (~10% as compared to uninfected) at 4 dpi when compared to the unchallenged control hamsters (*Figure 1A*; *Chan et al., 2020*; *Imai et al., 2020*; *Lee et al., 2020*; *Sia et al., 2020*). This data was accompanied by gross lung morphological changes characterized by pneumonitis regions with a high lung viral copy number and TCID$_{50}$ at 2 dpi that gradually decreased with disease progression, marking the recovery of the animals from SARS-CoV-2 infection (*Figure 1B and C*). COVID-19 is marked by severe pneumonia, lung injury, and an influx of activated immune cells (*Afrin et al., 2020*; *Boudewijns et al., 2020*; *Chen and Li, 2020*; *Moore and June, 2020*). In line with this, the histological analysis indicated an elevated disease index score marked by high scores of inflammation, epithelial injury, lung injury, and pneumonitis with elevated infiltration of granulocytes and mast cells on 2, 4, and 7 dpi (*Figure 1D and E*). The experimental control for toluidine blue showed negligible stain for mast cells (*Figure 1—figure supplement 1J*). Consistently, mRNA expression of mast cell signature enzymes, chymase and tryptase (*Caughey, 2007*), was increased on 2 and 4 dpi corroborating with mast cell enrichment (*Figure 1F*). Furthermore, the expression of eotaxin, a CC chemokine and a potent chemoattractant for airway eosinophils and mast cells found to be increased in asthma and allergy conditions, was upregulated at 2 dpi (*Figure 1G*; *Guo et al., 2001*). The expression of other lung injury markers like mucin (muc-1: marker of respiratory infections), surfactant protein-D (sftp-D: acute lung injury marker), advanced glycation end product (AGER: pro-inflammatory pattern recognition receptor), and plasminogen activator inhibitor 1 (PAI-1: a key factor for lung fibrosis) was upregulated at 4 dpi (*Figure 1G*; *Chatterjee et al., 2020*; *Crouch, 2000*; *Oczypok et al., 2017*; *Prabhakaran et al., 2003*). Collectively, our data show that SARS-CoV-2 infection in hamsters resembles human COVID-19 pathologies with a high viral load at the onset of infection marked by lung injury and inflammation.

## Early phase of SARS-CoV-2 infection in hamsters leads to an acute inflammatory response

Lung injury and multiple-organ failure have been attributed to the acute release of inflammatory cytokines caused by SARS-CoV-2 infection in COVID-19 patients (*Iwasaki and Yang, 2020*; *Mathew et al., 2020*; *Moore and June, 2020*; *Verity et al., 2020*). Most of the SARS-CoV-2 infection studies in hamsters have demonstrated the pathological changes associated with the dissemination of virus in various organs without characterization of an acute inflammatory response. We observed a profound splenomegaly in hamsters challenged with SARS-CoV-2 with an increase in spleen mass to body mass ratio at 2 and 4 dpi as compared to the uninfected animals (*Figure 2A and B*). As hamsters recover from SARS- CoV-2 infection indicated by gain of body weight, splenomegaly also subsided at the late phase of infection (7 and 14 dpi), which is corroborated with recovery in lung pathologies (*Figure 2A*). These observations, together, indicated that splenomegaly could be used as an indicator of disease severity and cytokine release syndrome, and thus can be used as one of the key parameters to determine SARS-CoV-2 infection in hamsters. To characterize the humoral immune response against SARS-CoV-2 infection in hamsters, serum IgG titer against viral proteins, RBD, spike, and N protein, was determined. As compared to a low dose, a high dose of infection generates a higher IgG titer against RBD (*Figure 2—figure supplement 1A*). The RBD, spike (prefusion S2P), and N protein ELISA detected the corresponding antibodies in challenged animals as early as 7 dpi. The serum endpoint antibody titers against RBD, spike, and N protein were 1:3610, 1:405, and 1:330,

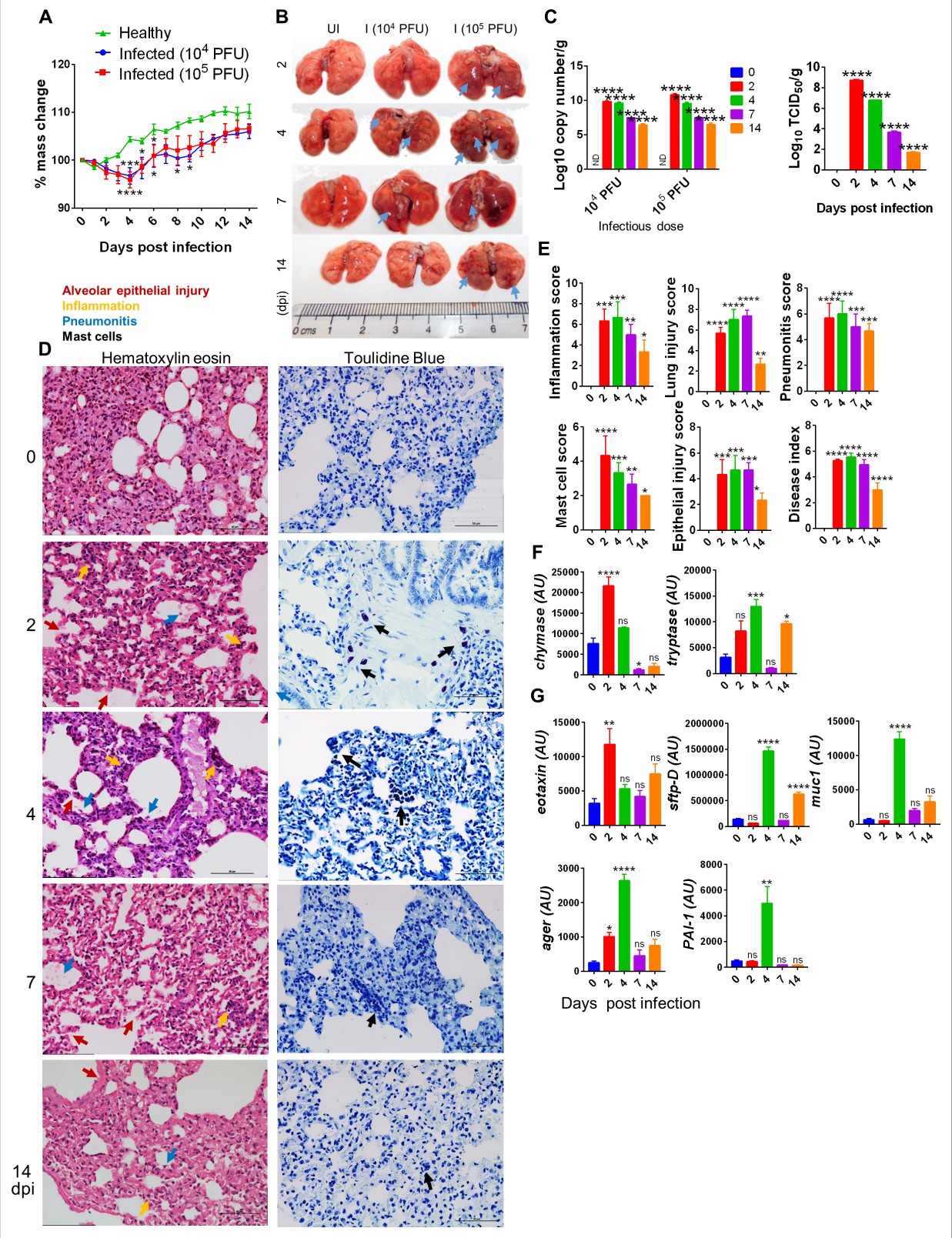

**Figure 1.** Pulmonary pathologies of severe acute respiratory syndrome coronavirus-2 (SARS-CoV-2)-infected hamsters. Hamsters infected with $10^4$ or $10^5$ plaque-forming unit (PFU) SARS-CoV-2 and the course of infection were followed for 14 days with the evaluation of clinical and pathological features of SARS-CoV-2 infection. (**A**) % body mass change. (**B**) Gross morphology of lungs showing pneumonitis regions and inflammation. (**C**) Lung viral load expressed as $\log_{10}$ copy number/g or $\log_{10}$ TCID$_{50}$/g. (**D**) Microscopic images of hematoxylin and eosin (HE)- and toluidine blue (TB)-stained lungs.

*Figure 1 continued on next page*

*Figure 1 continued*

(**E**) Blinded histological scores for pathological parameters on the scale of 0–5, where 0 indicates very low or no pathology while 5 indicates highest pathological change. Mast cells in TB-stained sections appear as dark blue-stained cells. (**F, G**) Lung qPCR showing mean relative mRNA expression with standard error of the mean (SEM). Alveolar epithelial injury (red arrow), inflammation (blue arrow), pneumonitis (yellow arrow), and mast cells (black arrow). *p<0.05, **p<0.01, ***p<0.001, ****p<0.0001 (one-way ANOVA, nonparametric Kruskal–Wallis test for multiple comparison).

The online version of this article includes the following figure supplement(s) for figure 1:

**Figure supplement 1.** Computational analysis deciphering the structural and interaction level insights of key host proteins, angiotensin-converting enzyme 2 (ACE2) (**A–E**) and neuropilin-1 (NRP-1) (**F, G**) involved in severe acute respiratory syndrome coronavirus-2 (SARS-CoV-2) infection.

respectively, and the titer of these antibodies was further enhanced to 1:8900, 1:550, and 1:660 at 14 dpi for PFU SARS-CoV-2 infection (*Figure 2C*). The cellular response was corroborated with the antibody response as the frequency of RBD-specific CD4$^+$IFNγ$^+$ cells was also found to be elevated (~2.5-fold) (*Figure 2D*). The frequency of CD4$^+$ T cells remained the same at 2 dpi between uninfected and SARS-CoV-2-infected hamsters (*Figure 2—figure supplement 1B*). To define the cellular response, particularly the T helper (Th) cell response, against SARS-CoV-2 infection in hamsters, we performed mRNA expression of crucial cytokines, chemokines, transcription factors, and checkpoint inhibitors. Our data demonstrated an increased mRNA expression of signature cytokines of Th1 (IFNγ, tumor necrosis factor [TNF]-α], Th2(interleukin [IL]-13), Th9 (IL-9), and Th17 (IL-17A) cells and IL-6 cytokine at the peak of infection (i.e., 2 dpi) along with their respective transcription factors, T-bet, GATA3, and RAR-related orphan receptor (ROR)-c (*de Candia et al., 2021*; *Grifoni et al., 2020*; *Mathew et al., 2020*). However, the transcription factor forkhead box protein (Foxp)-3 was found to be elevated at 14 dpi, suggesting a suppressive immunological response induced by regulatory T (Tregs) cells was increased with the declining of the infection. Interestingly, anti-inflammatory cytokines, IL-10, and transforming growth factor (TGF)-β, set in much early at 2 dpi, which could serve to counterbalance an aggressive inflammatory reaction (*Figure 2E and F*). We also found elevated expression of inducible nitric oxide synthase (iNOS) at 2 dpi compared to uninfected hamsters, indicating a possible role of an oxidative environment in SARS-CoV-2-induced pathologies (*Figure 2—figure supplement 1C*). Immune activation and infiltration at the target site is primarily dependent on chemokines and chemokine receptors. Therefore, we tested the expression of both chemokines and their receptors in SARS-CoV-2-infected hamsters. We found elevated expression of C-C chemokine receptor (CCR5) and C-C motif chemokine ligand (CCL)-5 at 2 dpi, which are known to regulate T cell function and chemotaxis. CCL-22, which is essential for Treg-dendritic cell (DC) cross-talk, was found to be elevated at 7 dpi as compared to uninfected hamsters. The expression of CXCL9 and CXCL-10, which are crucial for effector T cell trafficking and activation, and have been described as one of the biomarkers for CVC, was upregulated at 2 dpi as compared to uninfected hamsters (*Figure 2G*; *Altara et al., 2016*; *Hueso et al., 2018*). In addition, mRNA expression of PD-1 and its ligand, PDL-1, was found to be upregulated at the onset of infection, suggesting the induction of checkpoints to suppress the acute immune response induced by SARS-CoV-2 infection (*Figure 2H*). These data demonstrated that SARS-CoV-2 infection in hamsters induces an acute immune activation, chemotaxis, and expansion of immune cell populations at the peak of infection that resembles the cytokine storm patterns reported in symptomatic COVID-19 patients (*Afrin et al., 2020*; *Moore and June, 2020*).

## Characterization of cardiovascular pathologies in SARS-CoV-2-infected hamster

Myocardial injury is marked in around 25% of hospitalized COVID-19 patients, which includes thromboembolic diseases and cases of arrhythmia (*Giustino et al., 2020*; *Guo et al., 2020*; *Nishiga et al., 2020*). The interplay between host-virus interaction via ACE2-RBD and its impact on the host renin-angiotensin system (RAS) and immunological response is central to the development of CVC (*Giustino et al., 2020*). However, the lack of a suitable SARS-CoV-2 animal model has remained a limitation for studying cardiovascular-related complications associated with SARS-CoV-2 infection. Acute inflammatory responses like upregulation of CXCL9/10 and oxidative stress allowed us to rationalize that SARS-CoV-2 infection in hamsters may lead to CVC. Indeed, 7 and 14 dpi hearts showed clear ventricular hypertrophy, marked by a significant increase in ventricular mass to body mass ratio in challenged hamsters as compared to unchallenged hamsters (*Figure 3—figure supplement 1A and B*). It was further observed that the ventricular space was reduced as characterized by thickening of the

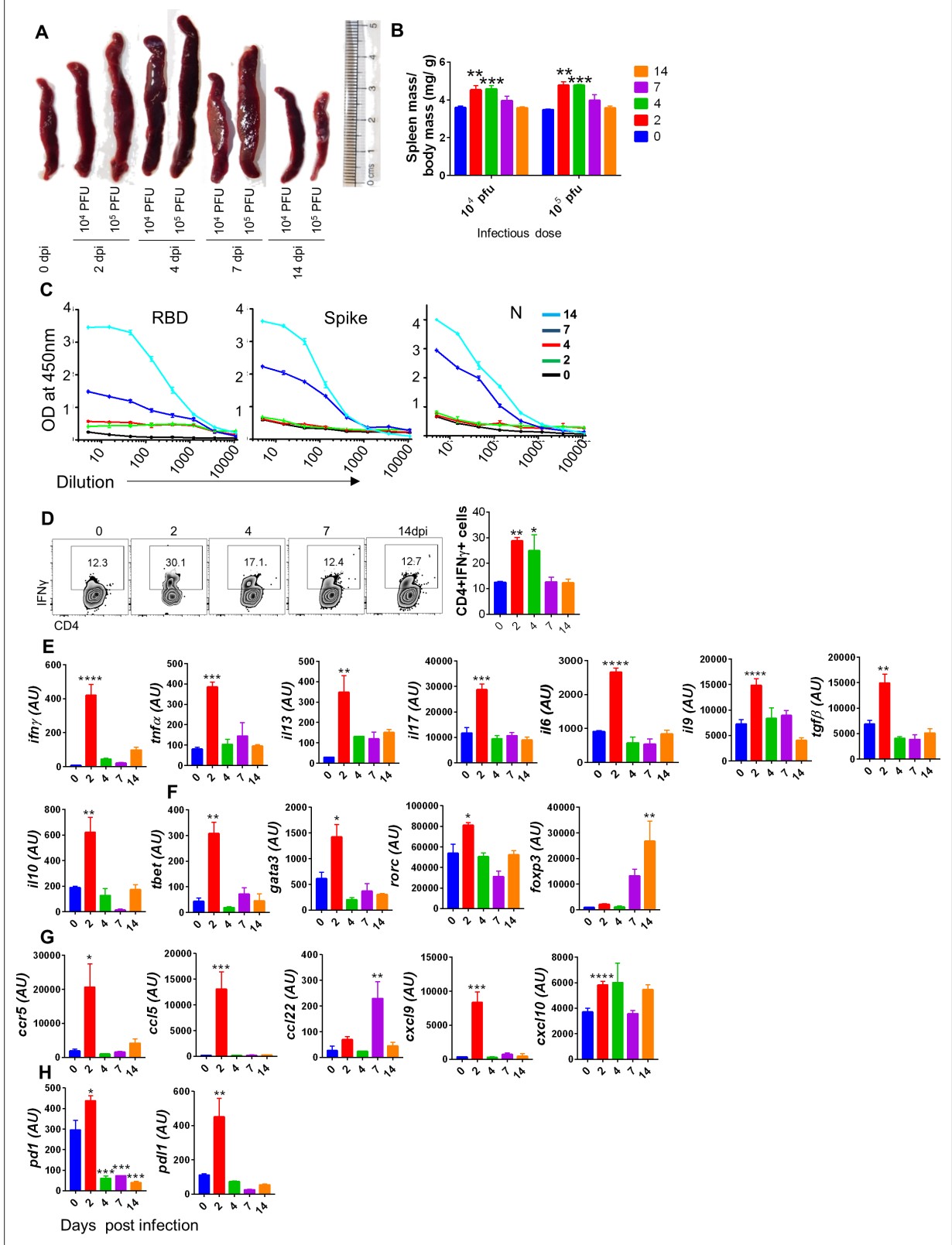

**Figure 2.** Immunological response against severe acute respiratory syndrome coronavirus-2 (SARS-CoV-2) infection in hamsters. Immunological responses of SARS-CoV-2-challenged hamsters were evaluated from serum or spleen samples. (**A**) Changes in spleen length of SARS-CoV-2-infected animals at different days post infection (dpi) as compared to uninfected animals. (**B**) Spleen mass to body mass ratio for excised spleen at different time points. (**C**) Serum IgG titer against SARS-CoV-2 viral proteins. (**D**) FACS for IFNγ secretion showing representative dot plots with % age frequency

*Figure 2 continued on next page*

*Figure 2 continued*

values and a bar graph showing mean ± SEM. Relative mRNA expression of (**E**) cytokines, (**F**) transcription factors, (**G**) chemokines, and (**H**) checkpoint inhibitors in the spleen of infected vs. uninfected hamsters. *p<0.05, **p<0.01, ***p<0.001, ****p<0.0001 (one-way ANOVA, nonparametric Kruskal–Wallis test for multiple comparison).

The online version of this article includes the following figure supplement(s) for figure 2:

**Figure supplement 1.** Immunopathological changes associated with severe acute respiratory syndrome coronavirus-2 (SARS-CoV-2) infection in hamsters.

ventricular wall and interventricular septum at 7 and 14 dpi in SARS-CoV-2-challenged hamsters as compared to uninfected controls (*Figure 3A and D*). The ventricular wall thickening at 7 and 14 dpi with SARS-CoV-2 infection in the hamster was marked by increased inflammation surrounding the coronary artery and elevated interstitial coronary fibrosis (*Figure 3B–D*). The experimental control Masson's trichrome (MT) staining of the heart showed staining pattern comparable to the uninfected heart (*Figure 3—figure supplement 1C*). Interestingly, increased fibrosis is an established pathophysiological stage in the majority of CVC (*Kong et al., 2014*; *Travers et al., 2016*). To further validate the CVC pathology observed during the late phase of SARS-CoV-2 infection in hamsters, we evaluated the CVC biomarkers in serum and viral load in the heart. There was a gradual increase in serum cardiac troponin I (cTnI) levels from the early to late phase of SARS-CoV-2 infection in hamsters (*Figure 3E*). Remarkably, remdesivir, which has been shown to have significant antiviral efficacy in hamsters, was able to inhibit serum cTnI levels at 4 dpi when compared to the infection control (*Figure 3—figure supplement 1D*; *Cortinovis et al., 2021*). We also found elevated serum IL-6 levels at 2 dpi (*Figure 3F*). In addition to inflammation indicator, IL-6 is used as a prognosis marker for CVC. Our data indicated that SARS-CoV-2 infection increases serum levels of IL-6 ,which decreases during later time points of infection (*Figure 3F*, *Figure 3—figure supplement 1E*). The elevated levels of CVC markers were accompanied by a high viral load in the heart of infected hamsters at 2 dpi as compared to uninfected controls (*Figure 3G*). The viral load in the heart was significantly reduced with remdesivir administration and was in line with reduced cTnI levels during infection (*Figure 3—figure supplement 1F*). Interestingly, we did not find any significant viral load in the heart at 7 and 14 dpi. Together, our results provide evidence for CVC in hamsters during the later phase of SARS-CoV-2 infection.

## Long-chain fatty acid accumulation in SARS-CoV-2 infection hamsters may be responsible for CVC

Since CVCs are often linked with a perturbed serum lipid profile, we reasoned that cardiovascular pathologies of SARS-CoV-2 in hamsters may be related to changes in circulating lipid molecules (*Bruzzone et al., 2020*; *Wu et al., 2020a*). Indeed, the serum lipid profile showed elevated cholesterol, TGs, HDLs, LDLs, and very low-density lipoprotein (VLDL) levels at 4 dpi (*Figure 4A*). Since our preliminary data revealed CVC and perturbations in the lipid profile of SARS-CoV-2-infected hamsters, we set out to do a detailed characterization of modulation in the abundance of lipid moieties at different time points of infection through ultra-performance liquid chromatography-tandem mass spectrometry (UPLC-MS/MS) to identify potential lipid biomarkers associated with cardiovascular pathologies. Our LC-MS/MS data identified 831 lipid molecules from the hamster serum samples (*Figure 4B*). These 831 lipid molecules were from the following categories: 11 acylcarnitine (AcCar), 15 cholesterol ester (CE), 50 ceramide (CE), 2 co-enzyme Q (CoQ), 31 diglycerides (DG), 13 ditetradecanoyl-sn-glycero-3-phosphoethanolamine (DMPE), 10 hexocylceramide (Hex-Cer), 31 lysophosphatidylcholine (LPC), 19 lysophosphatidylethanolamine (LPE), 7 lysophosphatidylinositol (LPI), 105 oxidized (Ox), 107 phosphatidylcholine (PC), 41 phosphatidylethanolamine (PE), 7 phosphatidylglycerol (PG), 38 phosphatidylinositol (PI), 141 ether bond containing lipid (Plasmenyl), 4 phosphatidylserine (PS), 56 sphingomyelin (SM), 5 saturated lipids (SO), and 135 TGs (*Figure 4C*). In addition, the uniquely and significantly modulated serum lipid molecules in hamsters, at 2, 4, 7, and 14 dpi with SARS-CoV-2, were identified by carrying out t-test analysis followed by multiple testing correction as compared to uninfected (0 dpi) control. Volcano plots for 2, 4, 7, and 14 dpi (top to bottom) show significantly modulated lipid molecules in red dots. Interestingly, majority of the significantly modulated lipid molecules were identified at the late phase of infection, that is, 7 or 14 dpi serum samples corroborating with the appearance of CVC in 7 dpi hamster hearts (*Figure 4D*). In total, 275 lipids were differentially regulated

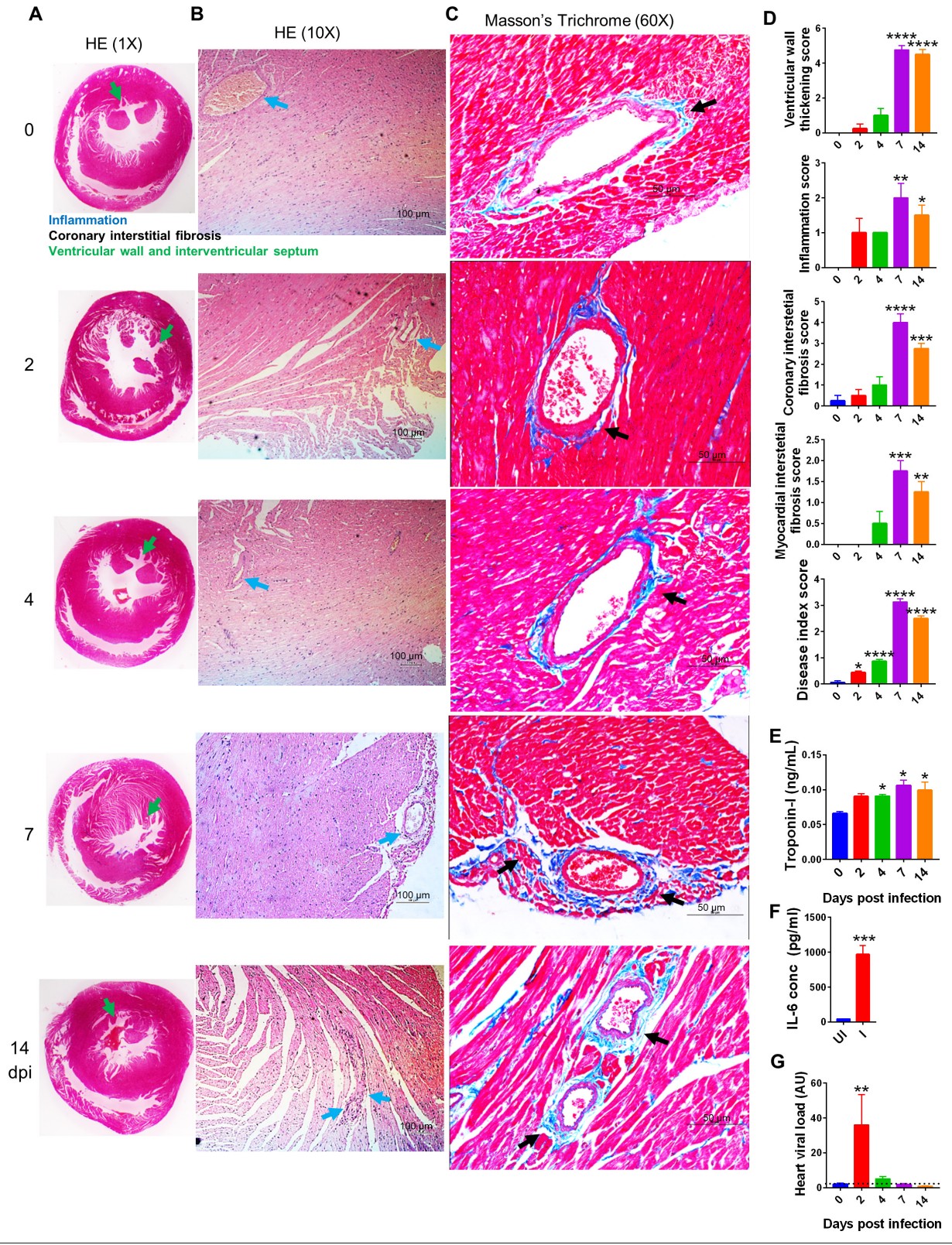

**Figure 3.** Cardiovascular complication (CVC) in hamsters challenged with severe acute respiratory syndrome coronavirus-2 (SARS-CoV-2). Histopathological changes in infected hamsters along with serum markers for CVC were evaluated. (**A**, **B**) Images of hematoxylin and eosin (HE)-stained heart captured at 1× and 10× showing ventricular walls and interstitial septum (green arrow) and inflammation around coronary artery (blue arrow). (**C**) Heart Masson's trichrome (MT) stains showing interstitial coronary fibrosis (black arrow). (**D**) Blinded histological scores for ventricular wall thickening,

*Figure 3 continued on next page*

*Figure 3 continued*

inflammation, coronary interstitial fibrosis, myocardial interstitial fibrosis, and disease index score were performed by a trained pathologist on the scale of 0–5, where 0 indicates very low or no pathology while 5 indicates highest pathological change. (**E**) Serum cardiac troponin I levels and (**F**) serum IL6 levels from Uninfected (UI) and Infected (I) samples as measured by sandwich ELISA (t-test, nonparametric Mann–Whitney test). (**G**) Relative heart viral load in heart homogenate. *p<0.05, **p<0.01, ***p<0.001, ****p<0.0001 (one-way ANOVA, nonparametric Kruskal–Wallis test for multiple comparison).

The online version of this article includes the following figure supplement(s) for figure 3:

**Figure supplement 1.** Pathological changes in hearts of severe acute respiratory syndrome coronavirus-2 (SARS-CoV-2)-infected hamsters.

as identified by *t*-test analysis (described in the 'Materials and methods' section), of which 65 lipids molecules were upregulated while 208 were downregulated as identified by $\log_2$ fold change (FC) in volcano plots (*Figure 4E*). In addition, the heatmap for $\log_2$ fold normalized relative abundance values for differentially regulated lipids showed that different classes of lipid molecules were distinctly downregulated or upregulated with the progression of disease (*Figure 4F and G*, respectively). Our lipidomics data show that unsaturated TGs (4–11 double bonds [db]), plasmanyl-PE (1–6 db), and plasmanyl-TGs (1–3 db) were uniquely upregulated during infection while saturated TGs (1–3 db), DGs, oxidized lipids, PC, PE, and PI were found to be downregulated during infection. However, both saturated TGs and oxidized lipids, which were implicated in CVC, were downregulated upon infection. Quite remarkably, we found LCFA TGs to be more abundant in the serum of infected hamsters as compared to uninfected hamsters. On the contrary, medium-chain fatty acid (MCFA) TGs were more abundant in uninfected hamsters as compared to infected hamsters. This corroborated with a previously published report that showed that increased LCFAs and decreased MCFAs are associated with increased risks of heart diseases (*Labarthe et al., 2008*). Taken together, these data demonstrated a distinct lipidomic profile upon SARS-CoV-2 infection in hamsters that is in line with the lipid profile observed in clinical cases of CVCs.

## Metabolomics changes in SARS-CoV-2 infection in hamsters

Emerging literature has suggested that metabolomic changes are associated with COVID-19 patients (*Bruzzone et al., 2020*; *Grassin-Delyle et al., 2021*; *Shen et al., 2020*). Here, we attempted to identify signature metabolites that could help in predicting the severity and progression of COVID-19. To find a correlation of metabolic signatures of COVID-19 in SARS-CoV-2-infected hamsters, we carried out serum metabolomics analysis of 2, 4, 7, and 14 dpi hamster serum samples and compared them with uninfected (0 dpi) controls. A schematic flow for the metabolomics analysis is shown in *Figure 5A*. *t*-test analysis followed by false discovery rate (FDR) correction identified 62 differentially modulated metabolites out of 334 metabolites in hamster serum samples. Out of these 62 differentially modulated metabolites, 50 serum metabolites were found to be upregulated while 12 metabolites were found to be downregulated during infection as compared to uninfected control hamster serum (described in the 'Materials and methods' section) (*Figure 5B and C*). Furthermore, pathway enrichment analysis revealed that arginine biosynthesis and arginine proline metabolism are the two profoundly modulated metabolic pathways in SARS-CoV-2-challenged hamsters (*Figure 5D*). The volcano plots indicating $\log_2$ FC in serum metabolite abundance of infected vs. uninfected control were exploited to identify uniquely modulated metabolites at different time points of infection (*Figure 5E*). The $\log_2$ fold values were further used to plot the heatmap of upregulated or downregulated metabolites. Quite interestingly, 2 dpi and 14 dpi serum samples showed a higher number of significantly modulated metabolites while 4 dpi serum samples showed no significantly modulated serum metabolites (*Figure 5F*). Collectively, SARS-CoV-2-challenged hamsters showed a significantly altered serum metabolomics profile as compared to uninfected hamsters and posed a possibility of identifying a potential metabolic biomarker for SARS-CoV-2 infection in hamsters.

## Identification of potential metabolic biomarkers for disease pathophysiology

Next, in order to understand the correlation of differentially regulated metabolites identified by comparing infected hamsters with severe and nonsevere COVID-19 patients and healthy serum metabolomic profile, we exploited the metabolomics data recently published by (*Figure 6A*; *Shen et al., 2020*). When compared to COVID-19 serum metabolomics profile data published by Bo Shen et al.,

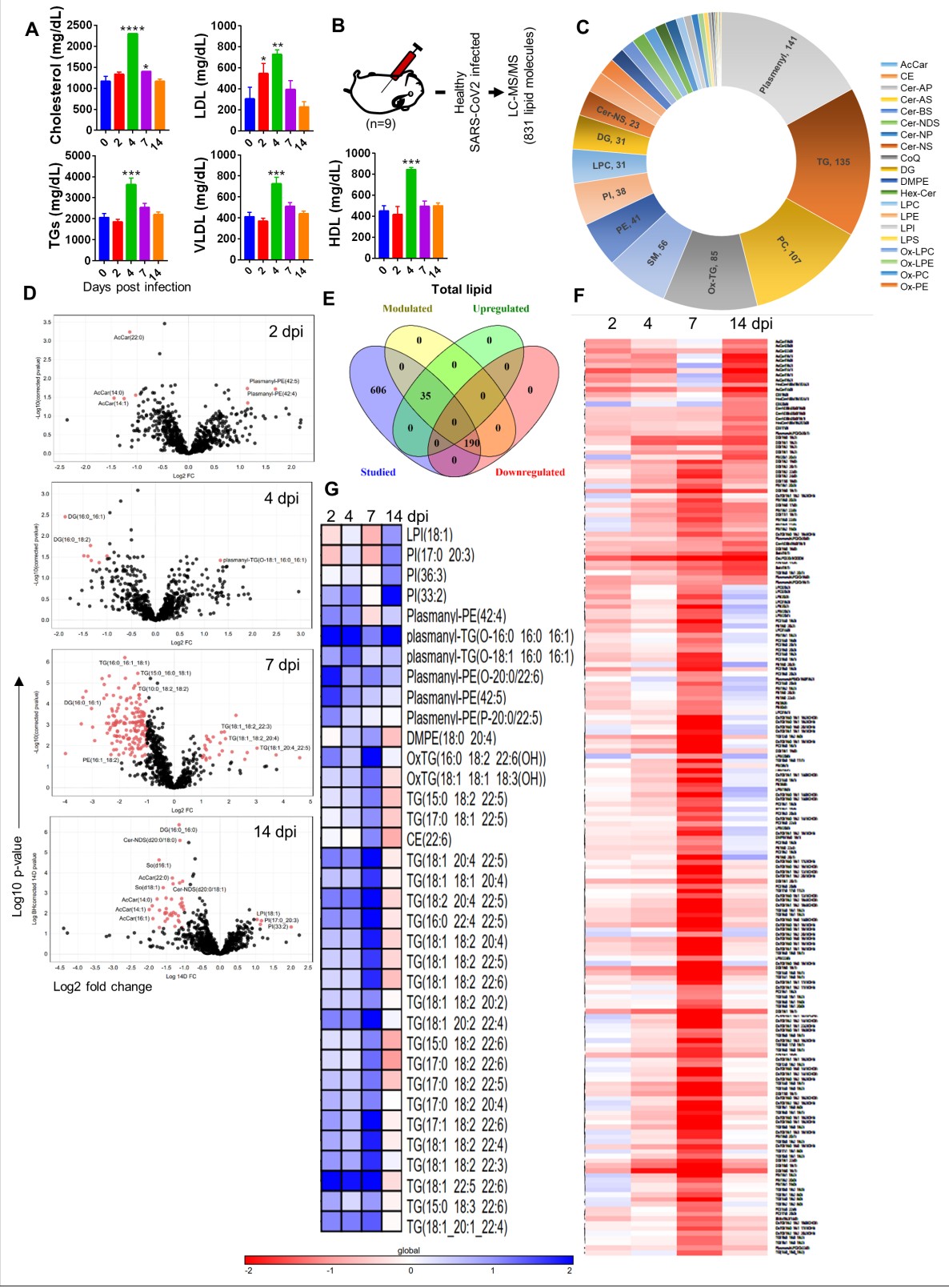

**Figure 4.** Serum lipid profile of severe acute respiratory syndrome coronavirus-2 (SARS-CoV-2)-infected hamsters. Lipid profiles from the serum samples of infected and uninfected hamsters were evaluated by biochemical and liquid chromatography-tandem mass spectrometry (LC/MS-MS) profiling. (**A**) Serum lipid profile measured by biochemical assay. (**B**) Schematic representation of lipidomics analysis from hamster serum samples. (**C**) Classification of detected lipid in this study. (**D**) Volcano plot showing log₂ fold change (x-axis) against -log₁₀ p-value (after false discovery rate [FDR]

*Figure 4 continued on next page*

*Figure 4 continued*

correction, Benjamini–Hochberg method) for 2, 4, 7, and 14 days post infection (dpi) vs. healthy control with significantly modulated lipids shown as red dots. (**E**) Venn diagram for differentially regulated lipids. (**F, G**) Heatmap showing $\log_2$ fold change of significantly modulated lipids at 2, 4, 7, and 14 dpi upregulated (**F**) or downregulated (**G**). *p<0.05, **p<0.01, ***p<0.001, ****p<0.0001 (one-way ANOVA, nonparametric Kruskal–Wallis test for multiple comparison). Panels (**F**) and (**G**) contain all lipids present significantly modulated in at least one or more time points.

we found six metabolites, viz, arginine, N-acetylglutamate, N-acetyl-tryptophan, ornithine, proline, and sphingomyelin, to be distinctly common between severe vs. healthy, nonsevere vs. healthy (Shen et al. study), and the hamster serum profile (***Figure 6B***). Furthermore, two metabolites were uniquely common between severe vs. healthy and hamster, that is, isovalerylglycine and uracil. Five metabolites were unique between the nonsevere and hamster profiles but were absent in the severe COVID-19 profile: 5-hydroxyindoleacetate, adenosine, carnitine, fumurate, and propionylcarnitine. Only one metabolite, N-acetylneuraminate, overlapped between severe vs. nonsevere COVID-19 profile and was also present in infected hamsters (***Figure 6B and C***). These common metabolites between hamsters and severe and nonsevere COVID-19 human serum samples prompted us to ask if these identified metabolites are also present in COVID-19 individuals in the early and late phases of infection. For this, we exploited our in-house COVID-19 serum metabolomics database generated from a longitudinal study of SARS-CoV-2-infected individuals (early phase = 17 subjects and late phase = 31 subjects) (***Figure 6D***). Among the 14 identified serum metabolites that were common between severe, nonsevere, and infected hamster serum profiles, it was found that only one, N-acetylneuraminate, was uniquely present in the serum of all the COVID-19 patients irrespective of disease pathology, disease stage, or demographic distribution, and was found significantly elevated in the SARS-CoV-2-infected hamsters (***Figure 6E***). Quite interestingly, N-acetylneuraminate had been previously described as CVC biomarkers and is correlated with increased heart risk (***Zhang et al., 2018***). Taken together, based on the previously published study, we here identify and propose N-acetylneuraminate as a reliable biomarker for SARS-CoV-2-infected hamsters.

In summary, we describe here pulmonary, immunological, and cardiovascular pathologies associated with SARS-CoV-2 infection in hamsters, establishing that SARS-CoV-2 infection in hamsters leads to CVCs in the later phase of infection (***Figure 7***). We further evaluated the molecular markers associated with CVC and found elevated levels of cTnI, cholesterol, LDLs, and VLDLs in the serum of infected hamsters. This was accompanied by an increased abundance of LCFAs as well as serum N-acetylneuraminate that was identified as a potential biomarker for SARS-CoV-2 infection in hamsters.

## Discussion

Pulmonary and extrapulmonary clinical and histopathological changes associated with SARS-CoV-2 infection in humans have been described by several groups (***Chen and Li, 2020***; ***Francis et al., 2021***; ***Lamers et al., 2020***; ***Mao et al., 2020***; ***Moore and June, 2020***; ***Neurath, 2020***; ***Nishiga et al., 2020***; ***Verity et al., 2020***; ***Wang et al., 2020***, p. 19). Moreover, hamsters as a preclinical SARS-CoV-2 infection model were recently shown to mimic viral entry and replication in a manner naturally similar to humans, and therefore hamsters are regarded as a suitable and natural model for SARS-CoV-2-challenged studies (***Chan et al., 2020***; ***Imai et al., 2020***; ***Lee et al., 2020***; ***Sia et al., 2020***). SARS-CoV-2 infection in hamsters causes lung pathologies, which resemble that of human lung pneumonitis, inflammation, and alveolar epithelial injury (***Boudewijns et al., 2020***, p. 2; ***Chan et al., 2020***; ***Lee et al., 2020***; ***Sia et al., 2020***). This is not surprising as huACE2 and haACE2 receptors have been shown to share major sequence homology (***Chan et al., 2020***). Such similarities between hACE2 and haACE2 strongly point to interaction with SARS-CoV-2 RBD protein with a similar trend of binding affinity. Our in silico data on haACE2 extends previous knowledge, providing greater insight into the interaction established between haACE2 and RBD protein. Moreover, the interacting residues and changes in interaction energy are strikingly similar between hamsters and humans, and as a result, the furin-cleaved S subunit could enter the interacting groove of haNRP-1 and prime it for interaction with ACE2 in a manner similar to that seen in humans. Our in silico data was suggestive of similarity in SARS-CoV-2 interaction in hamsters and humans, and it corroborated well with intranasal infection and subsequent lung viral load in hamsters as reported by previous studies. As increased viral load in the lungs is reflective of lung pathologies and has been shown to induce inflammation and injuries

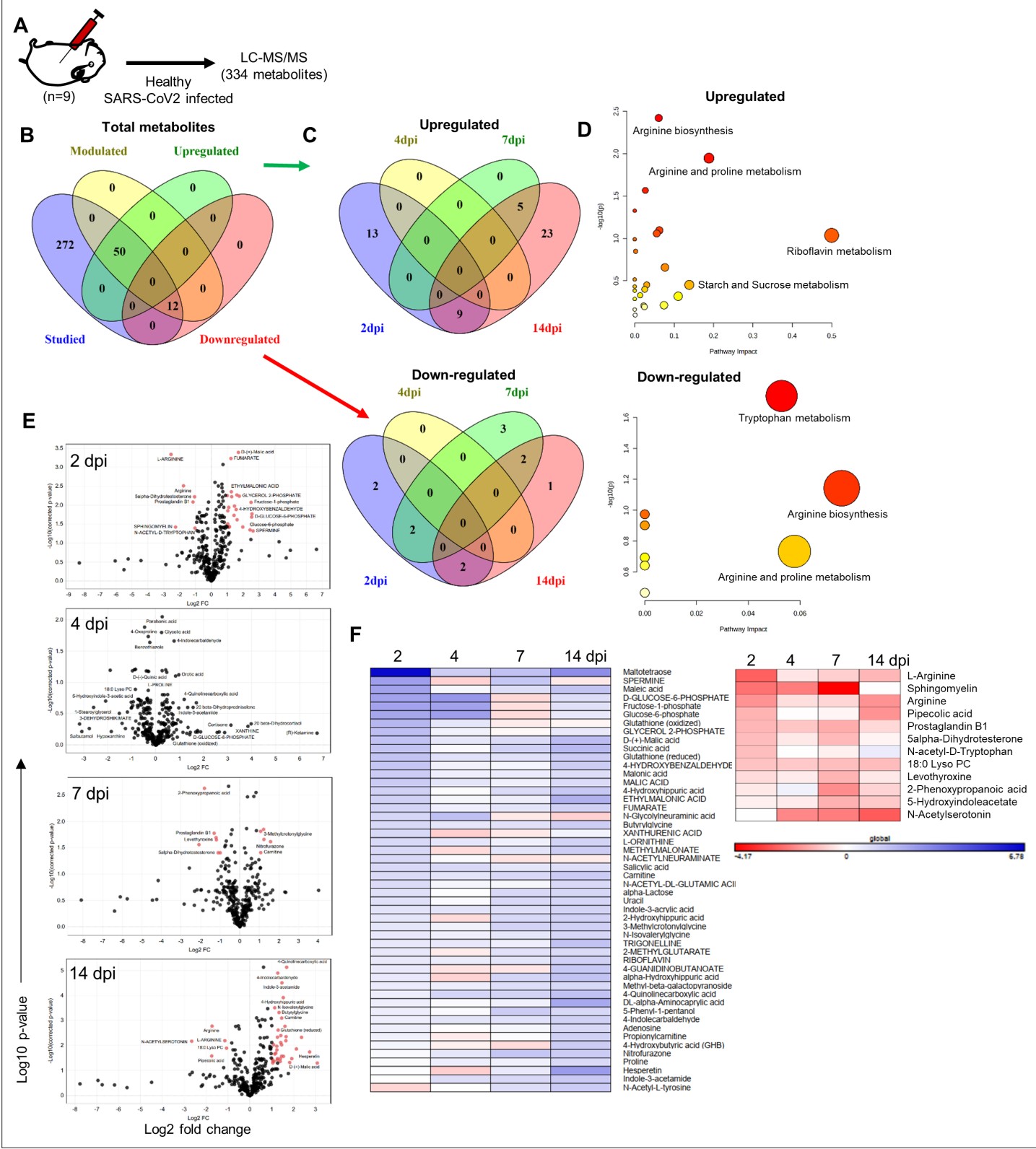

**Figure 5.** Serum metabolomics profile of severe acute respiratory syndrome coronavirus-2 (SARS-CoV-2)-infected hamsters. (**A**) Schematic representation of metabolomics analysis from the serum samples of infected and uninfected hamsters. (**B**) Venn diagrams for differentially regulated metabolites identified in this study classified as (**C**) upregulated and downregulated metabolites. (**D**) Metabolic pathways perturbed by changes in metabolomics profile. (**E**) Volcano plot showing log₂ fold change (x-axis) against -log₁₀ p-value (after false discovery rate [FDR] correction, Benjamini–

*Figure 5 continued on next page*

*Figure 5 continued*

Hochberg method) for 2, 4, 7, and 14 days post infection (dpi) vs. healthy control with significantly modulated metabolites shown as red dots.
(**F**) Heatmap for log$_2$ fold changes of significantly modulated serum metabolites at 2, 4, 7, and 14 dpi upregulated (left) or downregulated (right). Panel (**F**) contains all metabolites present significantly modulated in at least one or more time points.

in clinical cohorts, we reasoned that consistent pathophysiological changes may be occurring in the hamster as well (***Leng et al., 2020***). In line with this, we observed profound pneumonitis, inflammation, and alveolar epithelial injury upon histological assessments of the lung. Our data show that lung pathologies long persist even after a decrease in lung viral load since we observed significantly higher pathological scores on 7 dpi. This corroborates well with clinical cases of COVID-19 where it has been found that lung pathologies such as pneumonitis and lung injury persist longer even after no viral load was detected in the oral and nasal swabs.

Viremia is an important pathological characteristic of SARS-CoV-2 infection in humans, and further, SARS-CoV-2 is known to shed viral proteins that remain in circulation in the serum long after the virus is cleared, which provide humoral immunity against subsequent strain-specific infection (***Atkinson and Petersen, 2020***). The host responds to these pathogenic factors by producing IgG antibodies for effective neutralization (***Rogers et al., 2020***). We found that the IgG response to SARS-CoV-2 was most profound on 7 and 14 dpi, which corroborates well with recovery from SARS-CoV-2 infection in the hamster model. In addition to the anti-SARS-CoV-2 antibody response, a robust host response against SARS-CoV-2 depends on T cell proliferation and activation. Consistently, our data demonstrates profound splenomegaly in SARS-CoV-2-infected animals on 2 and 4 dpi with an increased spleen mass to body mass ratio, pointing to active immune cell stimulation and proliferation. Flow cytometry analysis indicated a significantly elevated IFN-γ response at 2 dpi hamsters, which could be one of the mechanisms of the antiviral response mounted during the early phase of SARS-CoV-2 infection. In addition, the Th17 signature cytokine IL-17A and its master transcription factor, RORc, were also found to be elevated in early phase of infection. Numerous clinical studies have shown that SARS-CoV-2 infection is characterized by a cytokine storm syndrome, which is responsible for lung pathologies and respiratory distress in severe COVID-19 cases. IFNγ, IL-6, and IL-17A were found to be important mediators and the hallmark of acute inflammatory response induced by SARS-CoV-2 infection (***Moore and June, 2020***). These cytokines were found to be significantly elevated in the early phase of SARS-CoV-2 infection in hamsters. Furthermore, the qPCR analysis revealed a similar activation pattern for several chemokines and inflammatory cytokines together with their respective transcription factors, indicating that immune activation sets in quite early during infection, and thus plays a role in pathological damage associated with the lungs and other organs during infection (***Francis et al., 2021***).

Emerging reports suggest that SARS-CoV-2 infection affects the cardiovascular system, which results in inflammation, endothelial activation, and microvascular thrombosis (***Giustino et al., 2020***; ***Guo et al., 2020***; ***Nishiga et al., 2020***). CVC is one of the comorbidities found to be associated with COVID-19 patients. Since there is a lack of knowledge about CVC arising from SARS-CoV-2 infection in hamsters, we carried out a detailed histological analysis to monitor the changes occurring in and around cardiomyocytes of the heart muscle. Interestingly, we found a profound thickening of the ventricular and interventricular septum with a marked depression in ventricular space capacity on 7 and 14 dpi heart samples. These changes were accompanied by the deposition of fibrous mass around the coronary artery and inflammation, which is believed to be one of the major causes of CVC and arrhythmia (***Travers et al., 2016***). Observation of cardiovascular complications on 7 and 14 dpi was a remarkable finding as it hinted strongly at extrapulmonary damages caused by SARS-CoV-2 infection. It may be possible that the virus invades the heart and causes direct infection and injury to cardiomyocytes or it may release soluble mediators in the blood that may cause CVC. Serum cTnI is an established prognosis marker for CVC and is associated with an increased risk of heart failure. We found that cTnI levels, in SARS-CoV-2-infected hamsters, were increased significantly especially in the late phase of infection, that is, 7 and 14 dpi. In addition, the qPCR analysis suggested an approximately 15-fold increase in heart viral load at 2 dpi. It seems, therefore, that CVC pathologies in SARS-CoV-2 infection are a consequence of both SARS-CoV-2 infection of heart cells and release of soluble factors (***Francis et al., 2021***), causing CVC. Consistently, remdesivir treatment reduced heat viral and profoundly decreased the serum cTnI levels.

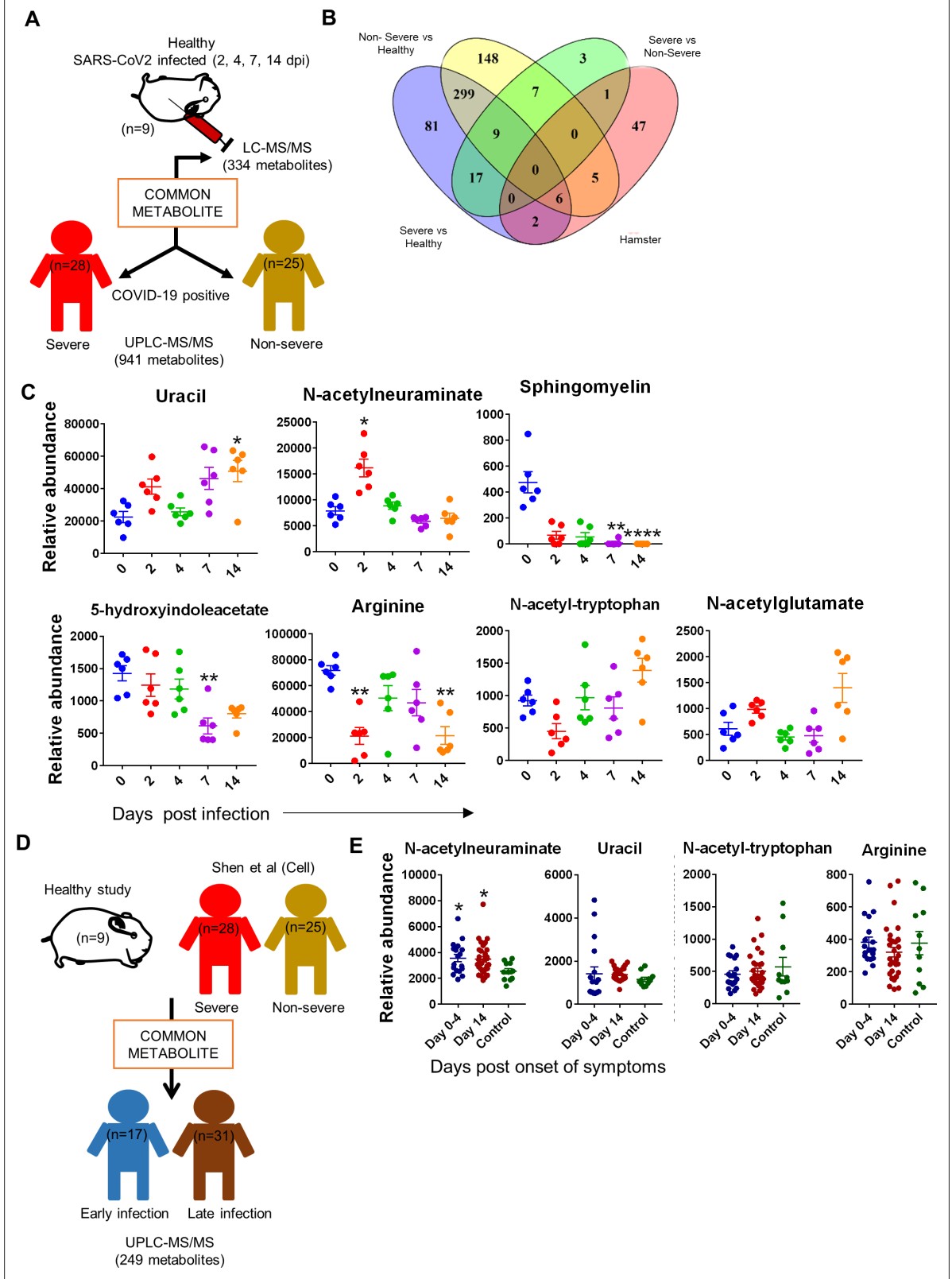

**Figure 6.** Identification of serum metabolic biomarker for severe acute respiratory syndrome coronavirus-2 (SARS-CoV-2)-infected hamsters. Serum metabolomics profile obtained from hamster study was then used to overlay with previously published and in-house-generated metabolomics profile database to identify serum metabolomics biomarker associated with infected hamsters. (**A**) Schematic representation of analysis methodology for comparative analysis between hamster and *Shen et al., 2020* database. (**B**) Venn diagram for correlation with *Shen et al., 2020* showing metabolites

*Figure 6 continued on next page*

*Figure 6 continued*

common between severe, nonsevere, and hamster study. (**C**) Relative abundance dot plot for metabolites common between severe, nonsevere, and hamster study. (**D**) Schematic representation of analysis methodology for comparative analysis between hamster, severe, nonsevere, early, and late phase infection metabolomics profile. (**E**) Dot plot for the relative abundance of metabolites common between hamster and human metabolomics profiles. *p<0.05, **p<0.01 (one-way ANOVA, nonparametric Kruskal–Wallis test for multiple comparison).

In addition to cTnI, we also characterized the serum lipid profile of SARS-CoV-2-infected hamsters. We found a dramatic increase in serum cholesterol, TGs, LDL, and VLDL levels in infected serum samples, which peaked at 4 dpi. Serum lipid molecules are other important soluble mediators that are associated with cardiovascular disease (*Labarthe et al., 2008*). We found that SARS-CoV-2 infection in hamsters leads to global changes in the lipid profile occurring at 7 and 14 dpi. In addition, LCFAs were found to be uniquely accumulated in serum samples of infected hamsters as compared to MCFAs during the late phase of infection, corroborating well with previously published reports showing that increase and decrease in LCFAs and MCFAs levels could be one of the biomarkers of CVC and heart failure.

In addition to lipid profiling, we attempted to evaluate the metabolomic changes occurring in hamsters infected with SARS-CoV-2 as there is no such data available on metabolic profiling in the hamsters. We identified 62 serum metabolites that were significantly modulated upon SARS-CoV-2 infection in hamsters. In order to identify signature metabolic biomarkers of SARS-CoV-2-infected hamsters that is also common for COVID-19 patients irrespective of the stage of infection, disease severity, or demography of infection, we followed a rigorous metabolomics comparative analysis between an already published Bo Shen et al. human database, our hamster study data, and our in-house-generated early and late phase human COVID-19 database. N-acetlyneuraminate was found to be a uniquely upregulated serum metabolic marker that was common across all these datasets. It is noteworthy that increased N-acetylneuraminate has been linked to increased heart disease risk (*Maciel et al., 2016*; *Zhang et al., 2018*). Together, our data provide the first insight into the key metabolites that could be used as biomarkers for the efficient and rapid evaluation of SARS-CoV-2 disease severity and could potentially be useful for testing the efficacies of antiviral drugs.

As of now, the hamster model for SARS-CoV-2 has been described as a suitable model to study lung pathologies and their associated clinical parameters (*Chan et al., 2020*; *Imai et al., 2020*; *Kaptein et al., 2020*; *Kreye et al., 2020*; *Lee et al., 2020*; *Osterrieder et al., 2020*; *Sia et al., 2020*; *Tostanoski et al., 2020*). However, a robust animal model that mimics extrapulmonary COVID pathologies, especially cardiovascular pathologies arising due to COVID-19, has not been evaluated. The current study provides a detailed insight into the cardiovascular and immunopathological changes associated with SARS-CoV-2 progression in hamsters. Our data further provides a detailed and comprehensive analysis of lipidometabolomic changes associated with SARS-CoV-2 infection in hamsters. Our data suggest that SARS-CoV-2 infection in hamsters can be classified into early and late phases of infection with distinct pulmonary and extrapulmonary pathophysiologies. While the early phase of infection is characterized by a high lung viral load, lung injury, and an acute inflammatory response, the late phase of the infection was characterized by the appearance of cardiovascular pathologies and heart-related complications. Together, our study provides the first proof of concept that hamsters could serve as a robust preclinical model for understanding pathologies associated with early and late phases of SARS-CoV-2 infection, and provide a basis to test therapeutics at different phases of SARS-CoV-2 infection-associated early and late pathologies.

## Materials and methods
### Animal ethics and biosafety statement
6–8-week-old female Golden Syrian hamsters were acclimatized in biosafety level-2 (BSL-2) for 1 week and then infected in Animal BSL3 (ABSL-3) institutional facility. The animals were maintained under 12 hr light and dark cycle and fed a standard pellet diet and water ad libitum. All experimental protocols involving the handling of virus cultures and animal infections were approved by RCGM, and the institutional biosafety and IAEC animal ethics committee.

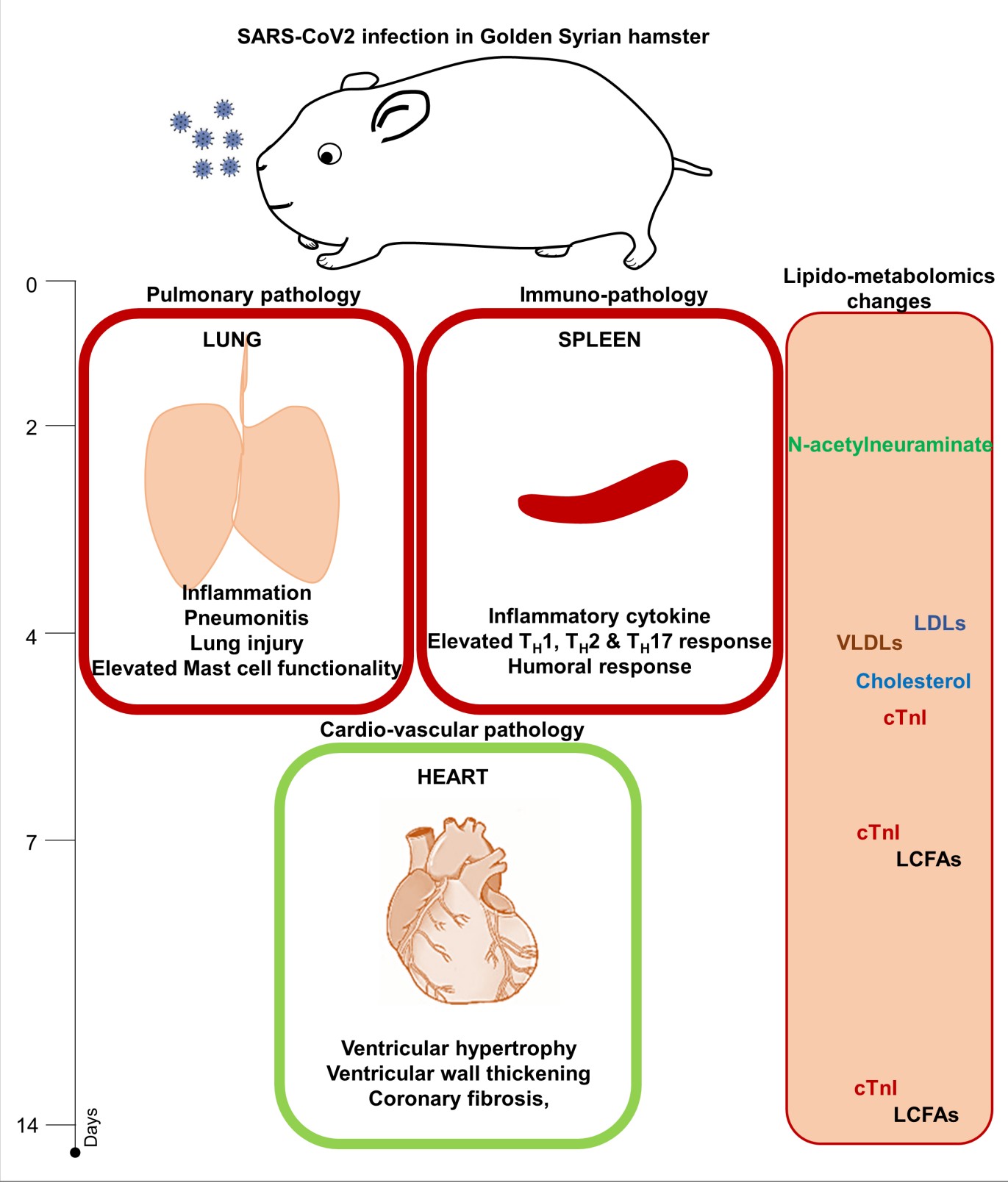

**Figure 7.** Summary Figure showing the study outline and important findings of the study.

## Virus preparation and determination of viral titers

SARS-related coronavirus 2, isolate USA-WA1/2020 virus was used as challenge strain and was grown and titrated in Vero E6 cell line grown in Dulbecco's modified Eagle medium (DMEM) complete media containing 4.5 g/l D-glucose, 100,000 U/l penicillin-streptomycin, 100 mg/l sodium pyruvate, 25 mM HEPES, and 2% FBS. The virus stocks were plaque purified and amplified at THSTI Infectious Disease Research Facility (biosafety level 3 facility) as described previously (*Harcourt et al., 2020*; *Mendoza et al., 2020*).

## SARS-CoV-2 infection in the Golden Syrian hamster

Infection in hamsters was carried out as previously described (*Chan et al., 2020*; *Sia et al., 2020*). The hamsters were briefly anesthetized through ketamine (150 mg/kg) and xylazine (10 mg/kg) intraperitoneal injections, and thereafter the infection was established intranasally with $10^4$ PFU (100 µl) or $10^5$ PFU (100 µl) of live SARS-CoV-2 or with DMEM mock control inside the ABSL3 facility. The remdesivir group received a subcutaneous (sc) injection of remdesivir at 15 mg/kg body weight 1 day before and 1 day post infection.

## Clinical parameters of SARS-CoV-2 infection

All infected animals were housed for 14 days, and their body mass was monitored as previously described (*Chan et al., 2020*; *Sia et al., 2020*). Nine animals from each group were sacrificed on 2, 4, 7, and 14 dpi, and their blood serum along with body organs such as lungs, intestine, liver, spleen, and kidney was collected. Serum samples were stored at –80°C until further use. Lung and spleen samples of infected and uninfected animals were compared for any gross morphological changes. Lungs samples were homogenized in 2 ml DMEM media and used for viral load and TCID$_{50}$ determination. A section of the lung and intestine along with other organs was fixed in 10% formalin solution and used for histological studies. The spleen was strained through 40 µm cell strainer with the help of a syringe plunger and used for qPCR and immunophenotyping studies.

## Viral load

Homogenized lung samples of 2, 4, 7, and 14 dpi along with uninfected controls were centrifuged for 10 min at 4°C, and their supernatant was collected. 100 µl of supernatant from each sample was then mixed with 900 µl of Trizol reagent (Invitrogen), and RNA isolation was carried out as per the manufacturer's protocol. Copy number estimation of SARS-CoV-2 RNA has been described previously (*Anantharaj et al., 2020*). Briefly, 200 ng of RNA was used as a template for the reverse transcription-polymerase chain reaction (RT-PCR). The CDC-approved commercial kit was used for of SARS-CoV-2 N gene: 5'-GACCCCAAAATCAGCGAAAT-3' (forward), 5'-TCTGGTTACTGCCAGTTGAATCTG-3' (reverse), and 5'-FAM-ACCCCGCATTACGTTTGGTGGACC-BHQ1 -3' (Probe) detection, and subgenomic RNA copy numbers were estimated by ΔΔCt method. Hypoxanthine-guanine phosphoribosyltransferase (HGPRT) gene was used as an endogenous control for normalization through quantitative RT-PCR. The region of N gene of SARS-CoV-2 starting from 28287 to 29230 was cloned into pGEM-T-Easy vector (Promega). This clone was linearized using SacII enzyme and in vitro transcribed using the SP6 RNA polymerase (Promega). The transcript was purified and used as a template for generating a standard curve to estimate the copy number of SARS-CoV-2 N RNA (*Anantharaj et al., 2020*).

## TCID$_{50}$

For TCID$_{50}$ determination, 50 µl of homogenized lung supernatant samples were incubated with confluent Vero-E6 cells in 96-well plates as described previously (*Chan et al., 2020*). Briefly, serial dilutions of 10-folds from each sample were added to the wells containing Vero-E6 cells monolayer in DMEM media in quadruplicate. After 4 days of incubation, TCID$_{50}$ was determined through the Reed and Münch endpoint method with one TCID$_{50}$ equivalent to the amount of virus required to cause a cytopathic effect in 50% of inoculated wells. TCID$_{50}$ values were expressed as TCID$_{50}$/g of lung mass. For the determination of virus titers, tissue samples (lungs) were homogenized to a final 10% (w/v) suspension in DMEM medium with gentamicin (Invitrogen, USA). The tissue samples from 2 to 14 dpi were used to infect Vero E6 cell monolayers in 48 plates as described previously. Virus titers were expressed as TCID$_{50}$/g of tissue.

## ELISA

The binding-antibody response to SARS CoV-2 post infection was measured using an ELISA-based platform as described earlier (*Sadhu et al., 2021*; *Shrivastava et al., 2018*). The antibody response was measured against spike, RBD, and N protein post-virus challenge at different time points. Soluble spike prefusion trimeric protein (S2P) (*Wrapp et al., 2020*), in-house RBD (*Parray et al., 2020*), and N proteins were coated on Maxisorp plates (Nunc) at different protein concentrations (spike S2P trimer; 1 µg/ml, RBD; 2 µg/ml and N protein at 0.5 µg/ml) in 1× carbonate/bicarbonate buffer, pH 9.6 overnight at 4°C. The following day, the plates were blocked using 250 µl of PBS containing 5% skimmed milk (blocking buffer). Threefold serially diluted sera (with 1:5 as starting dilution) in dilution buffer (1:5 times dilution of blocking buffer) were added to wells of the plates. The plates were incubated at room temperature (RT) for 1 hr and then washed three times with washing buffer (PBS + 0.05% Tween 20). The ELISA plates with N protein coating were washed additionally once with high salt PBST (phosphate buffer with 500 mM NaCl and 0.05% Tween 20) and incubated with biotinylated anti-hamster IgG antibody (Sigma) for another 1 hr and washed subsequently with washing buffer and incubated further with Avidin-HRP (Sigma) for 45 min at RT. Post-incubation plates were washed four times and 100 µl of TMB substrate (Thermo Fisher Scientific) was added to the washed wells. The reaction was stopped by adding 100 µl of 1 N $H_2SO_4$, and the plates were read at 450 nm on a 96-well microtiter plate reader. Sera endpoint titers were calculated as the reciprocal of serum dilution giving OD 450 nm readings higher than the lowest dilution of the placebo or control arm + 2× standard deviations.

IL-6 sandwich ELISA for hamster sera samples was performed according to the ELISA kit user manual (Immuno-tag, Cat# IT1441). Briefly, serum samples (diluted 1:3 times) along with a standard control were added to the precoated wells and then incubated with secondary antibody and streptavidin-HRP. Substrate was then added to each well after washing, and the reaction was stopped by using stop solution. The OD for each well was recorded as 450 nm against a blank control.

## In silico molecular modeling and protein-protein docking

In the absence of crystal structures of hamster ACE2, the crystal structure of human ACE2 and RBD, PDB ids 6m17 and 6VW1, was considered as templates to generate a robust hamster model through homology modeling (*Srivastava et al., 2018*). The robustness of each model was evaluated through ERRAT, PROSA, and Ramachandran plot (allowed and disallowed %), followed by structure optimization through energy minimization to allow conformational relaxation of the protein structure (*Parray et al., 2020*). The same was carried out for neurolipin (neuropilin)-1 protein of hamster as human neuropilin crystal structure (PDB id: 6JJQ) is reported. The most stable proteins were picked to perform the protein-protein (for RBD-ACE2) and protein-peptide (NRP-1 and CendR peptide of spike protein) docking studies through different algorithms. Rigid docking through PyDock allows some steric clashes while flexible docking by SwarmDock was done on relaxed structures of proteins to generate candidate solutions with at least one near-native structure (*Mattapally et al., 2018*). Clustering of docked poses was conducted to generate the most likely complex of ACE2-RBD and NRP1-Cend peptide based on the number of conformers and lowest binding energy (from –50 to –40 kcal/mol). An inventory of structural and energetic features of the complexes was obtained by analyzing the hydrogen bonds (cutoff of 3.5 Å for the donor-acceptor distance and 150° for the donor-hydrogen-acceptor angle) and hydrophobic contacts (nonpolar atoms separated by a distance of at most 4.0 Å). π-π interactions were considered to be formed when the short interatomic carbon-carbon distance was smaller than 4.8 Å (*Kanwal et al., 2016*). In an effort to dissect these interactions from the docking simulations, the total interaction energy between ACE2 and RBD was calculated within the framework of Amber force field description.

## qPCR

RNA was isolated from lung and heart homogenate and spleen samples using Trizol chloroform as previously described (*Rizvi et al., 2018*). Thereafter, RNA was quantitated by NanoDrop and 1 µg of total RNA was then reverse-transcribed to cDNA using the iScript cDNA synthesis kit (Bio-Rad; #1708891) (Roche). Diluted cDNAs (1:5) were used for qPCR by using KAPA SYBR FAST qPCR Master Mix (5X) Universal Kit (KK4600) on a Fast 7500 Dx real-time PCR system (Applied Biosystems), and the results were analyzed with SDS2.1 software. The relative expression of each gene was expressed

as fold change and was calculated by subtracting the cycling threshold (Ct) value of HGPRT (HGPRT-endogenous control gene) from the Ct value of the target gene (ΔCT). Fold change was then calculated according to the previously described formula POWER(2,-ΔCT) * 10,000 (*Malik et al., 2017*; *Rizvi et al., 2021*; *Roy et al., 2021*). The list of the primers is provided as follows:

| Gene | Forward | Reverse |
|---|---|---|
| HGPRT | GATAGATCCACTCCCATAACTG | TACCTTCAACAATCAAGACATTC |
| Rorc | GGGAAATGTGGGAACGCTGT | AAACGCAAGGCACTGAGGAA |
| Tryptase beta-2 | TCGCCACTGTATCCCCTGAA | CTAGGCACCCTTGACTTTGC |
| Chymase | ATGAACCACCCTCGGACACT | AGAAGGGGGCTTTGCATTCC |
| Muc1 | CGGAAGAACTATGGGCAGCT | GCCACTACTGGGTTGGTGTAAG |
| Foxo1 | AGGATAAGGGCGACAGCAAC | GTCCCCGGCTCTTAGCAAAT |
| Sftpd | TGAGCATGACAGACGTGGAC | GGCTTAGAACTCGCAGACGA |
| Eotaxin | ATGTGCTCTCAGGTCATCGC | TCCTCAGTTGTCCCCATCCT |
| Ager | ATTCCTGACGGCAAAGGGAC | ACTGTGTTCGAGGGACTGTG |
| PAI-1 | CCGTGGAACCAGAACGAGAT | ACCAGAATGAGGCGTGTCAG |
| IFNy | TGTTGCTCTGCCTCACTCAGG | AAGACGAGGTCCCCTCCATTC |
| TNFa | AGAATCCGGGCAGGTCTACT | TATCCCGGCAGCTTGTGTTT |
| IL13 | AAATGGCGGGTTCTGTGC | AATATCCTCTGGGTCTTGTAGATGG |
| IL17A | ATGTCCAAACACTGAGGCCAA | GCGAAGTGGATCTGTTGAGGT |
| IL9 | CTCTGCCCTGCTCTTTGGTT | CGAGGGTGGGTCATTCTTCA |
| TGFb | ACGGAGAAGAACTGCTGTG | GGTTGTGTTGGTTGTAGAGG |
| IL10 | GGTTGCCAAACCTTATCAGAA ATG | TTCACCTGTTCCACAGCCTTG |
| T-bet | ACAAGGGGGCTTCCAACAAT | CAGCTGAGTGATCTCGGCAT |
| GATA3 | GAAGGCAGGGAGTGTGTGAA | GTC TGA CAG TTC GCA CAG GA |
| FOXP3 | GGTCTTCGAGGAGCCAGAAGA | GCCTTGCCC TTC TCA TCC A |
| CCR5 | TGT GAC ATC CGT TCC CCC T | GGC AGG GTG CTG ACA TAC TA |
| CCL5 | CTACGCTCCTTCATC TGC CTC | CCT TCG GGT GAC AAA AAC GAC |
| CCL22 | CGT GGC TCT CAT CCT TCT TGC | CAG ATG CTG TCT TCC ACG TTG G |
| CXCL9 | TGG GTA TCA TCC TCC TGG AC | AAT GAG GAC CTG GAG CAA AC |
| CXCL10 | TGGAAATTATTCCTGCAAGTCA | GTG ATC GGC TTC TCT CTG GT |
| PD1 | CTGAAAAGGGGTTAAG CCA GC | GCC TCC AGG ATT CTC TCT GTT |
| PDL1 | TGATCATCCCAGACCCGCTC | CTC CTC GAA CTG CGT ATC GT |
| IL6 | GGACAATGACTATGTGTTGTTAGAA | AGGCAAATTTCCCAATTGTATCCAG |
| iNOS | TGAGCCACTGAGTTCTCCTAAGG | TCCTATTTCAACTCCAAGATGTTCTG |

## Histology

Excised tissues of animal organs were fixed in 10% formaline solution and processed for paraffin embedding. The paraffin blocks were cut into 3-μm-thick sections and then mounted on silane-coated glass slides. One section from each organ sample was stained with hematoxylin and eosin. Lung, heart, and colon samples were stained with toluidine blue, MT, and mucicarmine stains, respectively. Each stained section was analyzed and captured at ×10 and ×60 magnification. Heart images were also captured at ×1 magnification. Blinded assessment and scoring (on the scale of 0 [no observable pathological feature] to 5 [highest pathological feature]) of each sample section was performed by a trained pathologist.

## Immunophenotyping of splenocytes

0.5 million RBC lysed splenocytes were used for intracellular cytokine staining by restimulation with PMA (phorbol 12-myristate 13-acetate; 50 ng/ml; Sigma-Aldrich), ionomycin (1.0 µg/ml; Sigma-Aldrich), and monensin (#554724; GolgiStop, BD Biosciences) for 6 hr or with RBD protein (50 µg) for 72 hr. Cell surface staining with anti-mouse CD4 (GK1.5) PerCp (BioLegend) was carried out for 20 min in the dark at RT. Intracellular anti-mouse IFN-γ (XMG1.2) (BioLegend) staining was then carried out after fixing the cells in Cytofix solution and permeabilizing with 1X Perm/Wash Buffer using kit (BD Biosciences; #554714) for 20 min in dark at RT. Cells were then washed and acquired on FACS Canto II and were analyzed with FlowJo software (Tree Star) as previously described (*Malik et al., 2017*).

## Lipid extraction and lipidomics

Lipid extraction from serum samples was performed as previously described with some modifications (*Schwaiger et al., 2018*). 0.3 ml methanol was added to 100 µl of serum samples and mixed for 30 s followed by the addition of 1.25 ml methyl-tert-butyl ether. The mixture was incubated at RT for 1 hr on a shaker followed by the addition of 0.3 ml of MS grade water for the phase separation. 10 min after incubation at RT, samples were spun down at 400 rpm at 10°C for 5 min. The organic upper phase was collected and dried in a speed vac and stored at –80°C till further use. Moreover, the lipids extracted were dissolved in 100 µl of 65:30:5 (acetonitrile:2-propanol:water v/v/v). An acquity HSS T3 (2.1 mm × 100 mm × 1.8 Um, Waters) was then utilized for carrying out lipid separation by exploiting UPLC. Solvents A and B were water/acetonitrile (2:3 v/v) and 2-propanol/acetonitrile (9:1, v/v) respectively, at 0.3 ml/min flow rate and 40°C column temperature. 18 min total run time was utilized with the following gradient setup: 0–12 min solvent B ramped from 30% to 97% and hold for another 3 min. From 15.2 to 18 min, B was at 30%. Acquisition of data was carried out on a high-resolution mass spectrometer, Orbitrap Fusion (Thermo Fisher Scientific) equipped with a heated electrospray ionization source. Auxiliary gas and ESI sheath gas were 20 and 60, respectively. The negative and positive spray voltage was 3000 V. For a full MS run, 120k resolution was used with automatic gain control (AGC) targeted of 200,000 and mass ranges between 250 and 1200. The resolution was kept at 30K with AGC target 50,000 for Tandem mass spectrometry (MS/MS). Collision energy for fragmentation used was 27 ± 3.

## Lipid data analysis

LipidMatch Flow was utilized with default settings for peak picking (using mzmine), lipid annotation, blank filtration, and combining positive and negative data (*Koelmel et al., 2017*). Thereafter, data analysis was carried out as follows: the data was normalized by sum, Pareto scaled, and log-transformed for analysis in MetaboAnalyst.

## Metabolomics analysis of LC-MS/MS reverse phase and HILIC

Metabolites were extracted from 100 µl of serum samples by using 100% methanol. Thereafter, the mixture was vortexed for 1 min and kept on ice for protein precipitation. After centrifugation (10,000 rpm for 10 min at 4°C), the supernatant was collected and distributed in two tubes for polar and nonpolar metabolite analysis. The collected supernatants were then dried using a speed vacuum for 20–25 min at RT and stored at –80°C till further analysis. For the reverse phase, metabolites were dissolved in 15% methanol in water (v/v), and for polar metabolite analysis, samples were dissolved in 50% acetonitrile in water (v/v).

## Measurement of metabolites

Orbitrap Fusion mass spectrometer (Thermo Fisher Scientific) coupled with heated electrospray ion source was used for data acquisition. Data acquisition methods have been followed as per published protocols (*Kumar et al., 2020*; *Naz et al., 2017*) with minor modifications. Briefly for MS1 mode, mass resolution was kept at 120,000, and for MS2 acquisition, mass resolution was 30,000. Mass range of data acquisition was 60–900 Da. Extracted metabolites were separated on UHPLC UltiMate 3000. Data were acquired on both reverse phase and HILIC column and positive and negative ionization mode. The reverse phase column was HSS T3, and the HILIC column was XBridge BEH Amide (Waters Corporation). For polar compound separation, solvent A was 20 mM ammonium acetate in the water of pH 9.0 and mobile phase B was 100% acetonitrile. The elution gradient started from 85% B to 10%

B over 14 min with a flow rate of 0.35 ml/min. For the reverse phase, solvent A was water and solvent B was methanol with 0.1% formic acid added in both. The elution gradient started with 1% B to 95% B over 10 min with a flow rate of 0.3 ml/min. The sample injection volume was 5 µl. A pool quality control (QC) sample was run after every five samples to monitor signal variation and drift in mass error.

## Data processing

All LC/MS acquired data has been processed using the Progenesis QI for metabolomics (Waters Corporation) software using default settings. The untargeted workflow of Progenesis QI was used to perform retention time alignment, feature detection, deconvolution, and elemental composition prediction. Metascore plug of Progenesis QI has been used for the in-house library with accurate mass, fragmentation pattern, and retention time for database search. We have also used an online available spectral library for further confirmation of identification. The cutoff for retention time match was 0.5 min, and spectral similarity was more than 30% fragmentation match in Progenesis QI. Peaks that had a coefficient of variation (CV) less than 30% in pool QC samples were kept for the further analysis of data. Additionally, manual verification of each detected feature has been done for the selection of the right peaks.

## Lipidomics and metabolomics data analysis

For both metabolomics and lipidomics data, the data was cleaned up to remove zero intensity values. The average calculations were done with the remaining available intensity values. The fold change was calculated as the ratio of average intensity values of cases divided by the average intensity of controls as follows:

$$FC = \frac{\left(\frac{\sum_{i=1}^{m} I_{ki}}{m}\right)}{\left(\frac{\sum_{j=1}^{n} I_j}{n}\right)}$$

where FC is the fold change, $m$ is the number of available nonzero intensity values for case, $n$ is the number of available nonzero intensity values for control, $I_{ki}$ is the intensity of $i$th case sample for condition $k$, where $k \in$ {2D, 4D, 7D, 14D post infection}, and $I_j$ is the intensity of $j$th control sample.

A two-tailed $t$-test considering similar means of case and control population was used to calculate the p-values. The FC was transformed to $\log_2$ FC and a threshold was set to ±2 FC (±1 at $\log_2$ FC). The FDR was set at 0.05 for the identified lipids and metabolites. The significance threshold for differential expression was set using simultaneous threshold filters of the $\log_2$ FC of ±1 as well as a p-value threshold of ≤0.05. This was followed for both the metabolomics and the lipidomics data. The significantly expressed analytes (metabolites and lipids) are also represented on the volcano plots. For the volcano plots, the p-values were transformed into $-\log_{10}$ p-values for plotting. The significant values as described above were highlighted and labeled on the graph using Tableau (version 2020.3).

## Volcano plot

$-\log_{10}$ p-value vs. $\log_2$ FC was plotted on the volcano plot. Tableau (version 2020.3) was used to highlight and annotate some of the significant hits (metabolites/lipids). The points above or below ±1 $\log_2$ FC (i.e., ±2 FC) and 1.33 on the $-\log_{10}$ p-value scale (i.e., ≤0.05 p-value) were selected for annotation and were colored red for visual distinction. For the heatmap construction, we used $\log_2$ FCs for both the metabolites and lipids heatmaps. Gene-E software (https://software.broadinstitute.org/GENE-E/) was used to create the heatmaps. For the analytes (metabolite/lipid) with zero values in controls but a positive intensity reading value in cases, we considered them as overexpressed vs. control in that particular condition, and used a maximum $\log_2$ FC imputed manually. The value chosen for such cases was the maximum value from the data. For setting the color scale, we used blue to represent all values below $-4$ $\log_2$ FC (1/16 FC) and all values above $+4$ $\log_2$ FC were colored red (+16 FC).

### Whole data heatmap

ClustVis was used to generate the heatmap for the complete dataset for both metabolites and lipids.

## Principal component analysis (PCA) plots

Raw values of the metabolites and lipids were taken for all nine samples. The web-based tool ClustVis was used to create the sample PCA plots to check the clustering of biological samples (*Metsalu and Vilo, 2015*).

## Comparison against human data

The data from the article (*Shen et al., 2020*) was downloaded and manually compared with the hamster metabolites data obtained from our study. The metabolites common to each category were then compared against each other to identify metabolites unique to each category as classified by the authors.

### Human sera samples metabolomics

Human plasma samples were collected according to the recommended guidelines of the Institutional Ethics Committee (Human Research) of THSTI and ESIC Hospital, Faridabad (letter ref no: THS 1.8.1/ (97) dated July 7, 2020). Human blood samples were collected from COVID-19 patients and healthy individuals after the written informed consent. Individuals were enrolled in this study based on the inclusion/exclusion criteria prescribed by the Institutional Ethics Committee (Human Research) of THSTI.

## Patient characteristics

In the period between July 7 and September 4, 2020, we enrolled 31 COVID-19 patients that were asymptomatic or mildly symptomatic with COVID-19.

## Method

Blood samples were collected from RT-PCR-positive COVID-19-infected asymptomatic or mildly symptomatic individuals at 0–4 days post-onset of symptoms (POS) and 14 days POS. The blood samples were collected in Sodium Heparin CPT tubes. Similarly, blood from RT-PCR-negative, healthy individuals was collected in CPT tubes. The tubes were centrifuged at $1500 \times g$ for 25 min, and the plasma was carefully separated and aliquoted and stored at –80°C until further use.

### Statistical analysis

All the results were analyzed and plotted using GraphPad Prism 7.0 software. Body mass, lung mass, gene expression, FACS, ELISA, $TCID_{50}$, and qPCR studies were compared and analyzed using one-way ANOVA or Student's *t*-test. Metabolomics and lipidomics analyses were carried out with n = 9 serum samples per group. $p < 0.05$ was considered as statistically significant.

## Acknowledgements

Financial support was provided to the AA laboratory from THSTI core and Translational Research Program (TRP) and Department of Biotechnology (DBT) and DST-SERB. We acknowledge IDRF (THSTI) for the support at ABSL3 facility. Luvas (University of Hisar) and CDRI for providing the hamsters for the study. Small animal facility and Immunology Core for providing support in experimentation. We are garteful to Dr. Anil Kumar Pandey (ESIC, Faridabad) for providing Remdesivir for hamster expriments. ILBS for support in histological analysis and assessment. RCB microscopy facility for microscopic examination of the histology slide. We acknowledge the technical support of Upasna Madan. The following reagent was deposited by the Centers for Disease Control and Prevention and obtained through BEI Resources, NIAID, NIH: SARS Related Coronavirus 2, Isolate USA-WA1/2020, NR-52281. AKY was supported by DBT-Big Data Initiative grant (BT/PR16456/BID/7/624/2016) and the Translational Research Program (TRP) at THSTI funded by DBT.

## Additional information

### Funding

| Funder | Grant reference number | Author |
|---|---|---|
| THSTI core | | Amit Awasthi |
| Translational Research Program | | Amit Awasthi |
| DST-SERB | | Amit Awasthi |

The funders had no role in study design, data collection and interpretation, or the decision to submit the work for publication.

### Author contributions

Zaigham Abbas Rizvi, Data curation, Formal analysis, Investigation, Methodology, Visualization, Writing – original draft; Rajdeep Dalal, Srikanth Sadhu, Investigation, Methodology; Akshay Binayke, Sonu Kumar Gupta, Manas Ranjan Tripathy, Investigation; Jyotsna Dandotiya, Deepak Kumar Rathore, Amit Kumar Pandey, Methodology; Yashwant Kumar, Formal analysis, Investigation, Methodology, Resources, Software; Tripti Shrivastava, Guruprasad R Medigeshi, Sweety Samal, Investigation, Methodology, Resources; Suruchi Aggarwal, Methodology, Software, Validation; Amit Kumar Yadav, Data curation, Formal analysis, Investigation, Methodology, Resources, Software; Shailendra Asthana, Data curation, Formal analysis, Investigation, Methodology, Resources, Software, Validation; Amit Awasthi, Conceptualization, Data curation, Funding acquisition, Investigation, Methodology, Project administration, Resources, Software, Supervision, Validation, Writing – review and editing

### Author ORCIDs

Zaigham Abbas Rizvi (ID) http://orcid.org/0000-0001-7850-5672
Akshay Binayke (ID) http://orcid.org/0000-0002-9808-9036
Amit Kumar Yadav (ID) http://orcid.org/0000-0002-9445-8156
Amit Awasthi (ID) http://orcid.org/0000-0002-2563-1971

### Ethics

Human subjects: Human plasma samples were collected according to the recommended guidelines of the Institutional Ethics Committee (Human Research) of THSTI and ESIC Hospital, Faridabad (Letter Ref No: THS 1.8.1/ (97) dated 07th July 2020). Human blood samples were collected from COVID-19 patients and healthy individuals after the written informed consent. Individuals were enrolled in this study based on the inclusion/exclusion criteria prescribed by the Institutional Ethics Committee (Human Research) of THSTI.

This study was performed in accordance with the institutional animal ethics committee (IAEC) guidelines and all the protocols and procedures involved in the study were approved (IAEC approval number: IAEC/THSTI/94). The experimental procedures on animals were followed in strict accordance with the animal handling and usage guidelines by the IAEC and small animal facility, THSTI. Infection through intranasal route was performed under anesthesia to minimize pain.

### Decision letter and Author response

Decision letter https://doi.org/10.7554/eLife.73522.sa1
Author response https://doi.org/10.7554/eLife.73522.sa2

## Additional files

### Supplementary files
• Transparent reporting form

### Data availability
All data pertaining to the manuscript are made available in Dryad.

The following dataset was generated:

| Author(s) | Year | Dataset title | Dataset URL | Database and Identifier |
|-----------|------|---------------|-------------|-------------------------|
| Rizvi ZA, Dalal R, Sadhu S, Binayke A, Dandotiya J, Kumar Y, Shrivastava T, Gupta SK, Agarwal S, Tripathy MR, Rathore DK, Yadav AK, Medigeshi GR, Pandey AK, Samal S, Ashtana S, Awasthi A | 2021 | Immunological and cardio-vascular pathologies associated with SARS-CoV2 infection in golden Syrian hamsters | https://doi.org/10.5061/dryad.vhhmgqnvt | Dryad Digital Repository, 10.5061/dryad.vhhmgqnvt |

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
