## [Editor Report]

This study is of broad interest since it does provide data on cardiovascular disease in a hamster model of SARS-CoV-2 disease. It also makes interesting novel connections to lipidomic and metabolomic profiles in the context of the complications of a virally induced disease.

---

## [Decision Letter]

**Decision letter after peer review:**

[Editors’ note: the authors submitted for reconsideration following the decision after peer review. What follows is the decision letter after the first round of review.]

Thank you for choosing to send your work, "Immunological and cardio-vascular pathologies associated with SARS-CoV-2 infection in golden syrian hamster", for consideration at *eLife*. Your initial submission has been assessed by a Senior Editor in consultation with a member of the Board of Reviewing Editors. Although the work is of interest, we regret to inform you that the findings at this stage are too preliminary for further consideration at *eLife*.

There are many outstanding aspects of this work that the authors are commended for. The model clearly reproduces lung pathology attributed to this diseases and to this animal model, the immunological aspects are outstanding and the lipidomics and metabolomics-based studies are an outstanding and novel aspect, but on their own they do not yet tell a cohesive story or explain the histological findings that the authors need to stress..

The main area of weakness, which would have represented the potentially most significant and novel aspect of this work, relates to the cardiac histology findings. The quality of the images does not lend itself to the interpretations currently made and many helpful suggestions have been made by the reviewers that could help the authors pursue this issue and this generate a very high quality contribution to the field that will be of very wide interest. So overall the reviewers feel this work has tremendous potential

*Reviewer #1:*

In this study, the authors aim to show that hamsters may be a useful model of SARS CoV-2 long-term sequelae or "long COVID." The authors present useful data on many aspects of acute SARS CoV-2 in hamsters that are not well demonstrated in the literature such as gene expression of major regulators of inflammation, lipidomics, metabolomics, and antigen-specific CD4^+^ T cell responses which are important features of SARS CoV-2 in this model with translational potential if hamsters are used to test therapeutics aimed at preventing long COVID sequelae. However, the pathologic data that is presented to link both the lipidomics and metabolomics to extrapulmonary pathology suggestive of long COVID is weak. This study would greatly benefit from the addition of a treatment group or extended study timeline to support the claim that hamsters model key aspects of long COVID as seen in people. The lack of convincing pathology undermines the authors' model that lipidomic and metabolic changes are supportive of hamsters as a translational model of long COVID.

Overall-there is lack of description on scoring systems for histology. The methods state that pathology assessment was performed by a histologist (?)--pathology should be reviewed by a blinded veterinary pathologist. The data presented to show cardiac hypertrophy, fibrosis, and inflammation is not convincing. The pathology images are very low resolution and the fibrosis and inflammation cannot be observed clearly (if present). Control (sham infected, procedure-matched) and age matched tissue specimens are not shown. Organ weights for spleen and heart are not shown (preferably need to see organ/body weight measurements). Heart weight is the gold standard for cardiac hypertrophy-without this measure cardiac hypertrophy cannot be diagnosed; at a minimum measurement of the interventricular septum should be shown-the images do not convincingly show differences over time.

Ln 155: The images of mast cells are not clear-how was this scored (counts/high power field?). Please show control animals (not just time 0).

Likewise increased mucin staining in the GI tract is also not convincing. Please show controls over time.

*Reviewer #2:*

There are many published studies in which golden Syrian hamsters (GSH) are used as a small animal model for human SARS-CoV-2 infection and associated pathology. In this study, Rizvi, Zaigham Abbas et al. investigated several aspects of SARS-CoV-2 infection of GSHs, with an emphasis on the usefulness of the model to study cardiovascular complications in human COVID-19. From in silico analyses, they confirm with more detail than previous reports that hamster angiotensin converting enzyme-2 (ACE-2) interactions with SARS-CoV-2 spike protein are predicted to be very similar to human ACE-2 interactions, and they show novel data that predict hamster neuropilin-1 (NRP-1) interaction with spike protein are similar to human NRP-1 interactions. A shortcoming of this analysis is lack of biochemical or blocking studies, either in vivo or in vitro to establish of NP is an important receptor for viral infection of hamster cells. The authors histological and gene expression analysis of pulmonary pathology convincingly recapitulates the severe but resolving inflammatory lung disease in the hamsters that has been described in other publications, validating to relevance of the model in their hands before exploring extrapulmonary manifestations. The lung studies here also added some limited new findings that type 2 inflammation with mast cell activation and eotaxin are prominent in the hamster, which has also been implicated in some human studies.

Analyses of serum antibodies specific for viral RBD, spike and N proteins, spleen size, splenic CD4^+^ IFN-γ+ T cell numbers, and splenocyte mRNA expression all demonstrated a vigorous systemic immune/inflammatory responses, with most parameters peaking by day 2 after intranasal infection, and returning to baseline by day 14, while the serum anti-viral IgG titers showed continuous elevation from d2 to d14. Although the interrogation of immunological responses was limited in scope, especially with respect to local tissue responses, and lack of single cell omic analyses, the results are sufficient to establish the similarity of the post infection disease in the authors hands with previous hamster reports and to compare with comparable human parameters, again validating the relevance of looking at metabolic and cardiovascular parameters not previously explored.

A major strength of this study is the intent to study cardiovascular complications in the hamster model, and to this end the authors focused on heart histopathology and serum lipids that are known to associate with human cardiovascular disease. The analysis of heart tissue was limited to H&E and Mason Trichome stained sections, to assess morphologic changes in myocardium and intramyocardial blood vessels, myocardial inflammation, and fibrosis. The interpretation that there is myocardial hypertrophy, increased coronary fibrosis, and increased inflammation in infected mice at days 7 and 14 is questionable based on the images shown in Figure 4, which appear to represent one section of one heart for each time point. Quantitative image analyses of the histopathogical parameters, including the hearts from the 9 animals sacrificed at each time point would help is assessing the significance of the findings. The images shown suggest some perivascular fibrosis, even at day 0, and mild perivascular inflammation, although the resolution of the images limits interpretation. Additional data showing cardiac inflammation and myocardial dysfunction by other methods would be helpful, such as transcriptional analysis, immunohistochemistry, or flow-cytometry of cardiac cell suspensions. Each or all could help substantiate the authors conclusions. A relevant but unanswered question is if there is viral infection of cardiac cells in their experimental animals, as myocyte SARS-CoV-2 infection has been reported in a few recent papers. The acute changes in blood cholesterol, lipoproteins, and triglycerides only seen at day 4, shown in Figure 4, are unlikely to have any consequences on long term vascular pathology or any myocardial pathology that may be present at day 7 or 14. Extremely elevated cholesterol levels over many weeks can lead to arterial atherosclerosis in some experimental rodent models, rarely resulting in myocardial ischemic injury, but there is no evidence of atherosclerosis or ischemic myocardial injury in this study.

The extensive blood lipidomic and metabolic analyses on the hamsters at different time points after infection, and comparison of with lipid and metabolite changes associated with human COVD-19 patients, is a major strength and novel aspect of this study. The value of these types of data in developing biomarkers of disease severity and risk for long term sequalae to infection remains to be determined, but the findings here help establish the value of the hamster model for such purposes. Nonetheless, these data are descriptive, and the causal connection of changes in any of the lipids or metabolites with putative cardiovascular pathologies is tenuous.

Overall, the authors succeeded in showing how the hamster model can be useful to study systemic extra -pulmonic pathologies and systemic metabolic changes associated with SARS-CoV-2 infection. The data shown do not make a compelling case for the hamster model to study long term effects of infection outside the lung, given that the longest time point examined was d14, and the cardiac/vascular data were not robust. Furthermore, most of the cardiovascular complications that have been reported in COVD patients concern thrombosis, yet this report does not show any evidence of thrombotic complications in the hamsters.

The possibility that cardiac cells in experimental animals are infected by SARS-CoV-2 should be tested. Recent reports that cardiac myocytes can be infected with SARS-CoV-2, including hamster myocytes (Chen, S et al. SARS-CoV-2 Infected Cardiomyocytes Recruit Monocytes by Secreting CCL2. Res Sq. 2020 doi: 10.21203/rs.3.rs-94634/v1; Marchiano, S et al. SARS-CoV-2 Infects Human Pluripotent Stem Cell-Derived Cardiomyocytes, Impairing Electrical and Mechanical Function. Stem Cell Reports. 2021 Mar 9;16(3):478-492. doi: 10.1016/j.stemcr.2021.02.008.)

Cardiac hypertrophy is best assessed by heart weight, and a second best way is to do careful quantitative analysis of left ventricular wall thickness on cross sections from all animals in each group. The photographs of two hearts.

Myocardial histopathology must be assessed and reported in quantitative fashion.

In Figure S4 the images labelled Kidney are Liver, and vice versa. The kidney images are of poor resolution, and no glomeruli can made out in the d2 section. Sections from d7 and 14 should also be shown.

If there are activating anti-hamster CD3 reagents available to activate T cells, that would be a better (more selective, more physiologic) way for activating spleen T cells than PMA/ionomyicn.

*Reviewer #3:*

In this work the authors examine changes in the lungs and extrapulmonary organs in the setting of SARS-CoV-2 infection using the golden Syrian hamster model, with emphasis on cardiac changes. This is a well-established model to study SARS-CoV-2 infection. The authors also examine changes in splenic inflammatory related molecules, serum lipid molecules and serum metabolites.

The major strength of the work is the attempt to perform detailed analyses of the extrapulmonary organs combined with splenic inflammatory changes, metabolomics and lipidomics in the setting of acute SARS-CoV-2 infection in an animal model.

The major weakness of this work is that often the histologic changes being reported, particularly in the heart, are not well supported by the data provided. In addition, more rigorous statistical approaches could be employed, and there is no clear mechanism proposed to link the changes in the splenic inflammation, serum lipidomics, and serum metabolomics with the proposed cardiac changes.

1. There are numerous studies now on SARS-CoV-2 in the hamster. There should be much better effort relating this work to the published literature, particularly stressing what is novel and what is not. It is not clear if any of the pulmonary studies being reported here are actually novel.

2. The authors claim: "Strikingly, SARS-CoV2 infection in the hamster leads to significant interstitial coronary fibrosis on 7 and 14 dpi characterized by thickening of ventricular walls interventricular septum." and later: "The ventricular walls thickening at 7 and 14 dpi with SARS-CoV2 infection in GSH was marked by increased inflammation surrounding the coronary artery and elevated interstitial coronary fibrosis (Figures 4B, 4C)." However, the cardiac changes being reported are simply not supported by the images provided. Heart weights and wall thicknesses need to be measured and compared across the groups using statistics. Also, cardiac fibrosis and inflammatory cell infiltration should be quantified and comparted across the groups using statistics. Better images showing clear changes correlating with the quantitative analyses would be required.

3. The authors state "…we reasoned that cardio-vascular pathologies of SARS-CoV2 in GSH may be related to changes in circulating lipid molecules (Bruzzone et 240 al., 2020; D. Wu et al., 2020). Indeed, serum lipid profile showed elevated cholesterol, TGs,HDL, LDL and VLDL levels at 4 dpi (Figure 4D)." However, the changes in serum lipids were very transient and unlikely to rapidly cause cardiac fibrosis. There is no clear logical mechanism being present linking the serum lipid levels with the proposed cardiac changes.

4. For all of the studies performed including the lipidomics and metabolomic studies the authors simply us a P-value of less than 0.05 as being statistically significant with no adjustment for multiple assessments or consideration of FDR. A more sophisticated statical approach is needed.

5. The spleens need to be weighed and the weights compared across the groups using statistics.

6. There are frequent statements in the discussion that simply do not match the data shown. For example, the authors state: "Our data show that lung pathologies long persists after the virus clearance from the lung since we observed a high pathological score on 7 dpi." However, the authors own data show that the virus is still present at 7 dpi.

7. All figure panels need proper labels, particularly both axes of any plots.

8. In Figure S4, the degree of mucin staining in the images does not appear to match that shown in the bar graphs. The scale for the scale bars is not legible. For Panel B the images marked "liver" are not liver, and the images marked "kidney" are not kidney.

9. There should be some rational mechanism linking the splenic inflammatory cell changes, lipidomics and metabolomics with the proposed cardiac changes.

[Editors’ note: further revisions were suggested prior to acceptance, as described below.]

Thank you for resubmitting your work entitled "Golden Syrian hamster as a model to study cardiovascular complications associated with SARS-CoV2 infection" for further consideration by *eLife*. Your revised article has been evaluated by Balram Bhargava (Senior Editor) and a Reviewing Editor.

This report suggests that the Syrian hamster has the potential to be a good model to examine the metabolic changes following heart injury caused by SARS-CoV-2.

The manuscript has been improved but there are some remaining issues that need to be addressed, as outlined below:

1) Revise statistical analyses using non-parametric approaches as suggested.

2) Minimize or remove connections made to long COVID- the evidence is not strong enough that this is a model.

3) Include recent references as suggested.

4) Extensive editing of grammar and syntax is absolutely necessary.

5) Make MS readouts legible.

*Reviewer #1:*

This manuscript nicely highlights SARS CoV-2 effects on the heart in the Syrian hamster model, particularly from the virologic, transcriptional, and metabolic standpoint. The data clearly highlights metabolic changes that support the exploration of using the hamster model for cardiovascular complications of COVID. The data for histopathologic evidence for cardiovascular changes remain unconvincing at the timepoints shown. I believe a longer-term study following hamsters for greater than 14 days would be more biologically relevant for histopathologic changes. Regardless, the biochemical and immunologic data support the use of this model for studying cardiovascular changes.

The authors have addressed all the concerns raised. Unfortunately, the histopathologic data, although shown as significant via scoring, is still not biologically convincing. The heart weight to body weight index tracks for infected animals only. The ventricular width should have been done quantitatively and not semi quantitatively (scoring) to be convincing. The appropriate statistical comparison would be heart mass/body mass of infected to uninfected animals at each time point assayed. Also, the semiquantitative scoring of inflammation and fibrosis does not match up with what can be seen by eye. Lastly, the scoring metric for pathology should be clearly outlined in the figure legend (i.e. 1-5 minimal, mild, marked--for what features). What is the total score possible?

Regardless--the cytokine, lipid, and metabolic profiling and viral load data from the heart is very interesting and supports the further exploration of this model for studying cardiovascular effects of COVID.

There is still substantial copy editing needed for this manuscript. Many sentences do not make sense and grammar needs substantial improvement prior to publication.

*Reviewer #2:*

The authors have studied SARS-CoV-2 infection in golden hamsters. Evidence is provided for viral RNA in the heart and for many alterations in the circulation, discovered by metabolomics and lipidomics, including many that have been linked to cardiovascular disease in humans. The novel aspect of the studies presented are the studies utilizing lipidomics and metabolomics and the abrogation of heart disease after Remdesevir. This is a revised manuscript and most of the concerns of the previous reviewers have been largely addressed. However, there is at least one other relatively recent study that the authors have failed to make note of, (and that was published in July 2021), that has also observed myocardial inflammation and evidence for viral RNA in the heart. So, some of the findings have been independently confirmed and that is viewed as a strength rather than as a limitation. The other study made simple, direct links between the presence of viral RNA in a tissue and the events that lead to cardiac inflammation.

The authors have made some important improvements to the first submission especially in terms of better documenting the cardiac involvement in the model.

At this point in time changes like the increase in circulating sialic acids are interesting but most of the metabolic and lipidomic changes are best attributed to systemic inflammation as well as to heart damage. Given the time frame, it is difficult to consider these alterations as causal for the heart inflammation seen but these changes could be of relevance when one considers the cardiovascular complications of clinical COVID-19 that fall into the PASC spectrum.

The authors should more comprehensively reference earlier papers on the use of hamsters as a model for SARS-CoV-2 including studies in 2020 from Rosenke et al. (PMID 33251966) and Boudewijns et al. (PMID 33203860). However, they should, at least during this submission, have quoted a manuscript showing cardiovascular disease in a similar model that was published in July 2021. This study published about four months ago from Francis and colleagues. (PMID 34265022) should certainly have been discussed because it in part supports the authors' studies and demonstrated inflammation in the heart. In the studies of Francis et al., there was viral RNA in the heart though live virus was not recovered from this organ, and there was evidence provided for Type I interferon dysregulation, eosinophilic myocarditis, inflammatory cytokines, T cell infiltration and activation, though the evidence suggested the absence of TH2 cells and T regs.

There remain issues related to the authors' lack of care with syntax and grammar that still need to be taken care of and careful proofreading of the manuscript should be undertaken.

*Reviewer #3:*

– A central issue is that depending on the dose, SARS-CoV-2 is cleared in 5-6 days in the hamster model. Authors need to defend and define how can this is appropriate for modeling late phase effects of COVID-19 in humans.

– Throughout their figures the authors indicate that one-way ANOVA is used to compare the groups but do not specify the post-hoc test being used to compare the individual groups. E.g. Dunnett's test would be the most appropriate for comparing each group to the control (e.g. time zero). However, ANOVA with post-hoc should be applied only to parametric data (i.e. normally distributed measurement data). However, the scoring data will be non-parametric and will need to be compared using a non-parametric test.

– In their results (line 158), the authors equate recovery with elevated antibodies at day 14. Elevated is antibodies is expected (e.g. would get same result in vaccination) and should not be taken as a proxy for recovery.

– Lines 187-189. authors contend that a cytokine storm is in operation during acute infection. Circulating levels of IL-6 is one of the key cytokines that predicts storms in human COVID-19. Why was this not measured? IL-6 mRNA was measured later, but the authors should measure the actual level of IL-6 in circulation.

– It seems that in the biochemical analysis vs LC-MS/MS analysis paint different pictures regarding the time course of lipid biosynthesis. It seems that in the latter case, the authors are more interested in reporting changes, rather than the trajectory of those changes (e.g. in Figure 4G and F it's not possible to tell what is up and what is down). This paints a confusing picture. It also doesn't help that the text on Figure 4F is too small to read.

– Use Kruskal-Wallis test for comparing non-parametric score/count data.

– Would suggest disentangling the proposed parallels between late stage effects seen him COVID-19 in humans with SARS-CoV-2 infection in hamsters, unless longer term effects well outside the two week sampling window can be demonstrated.

---

## [Author Response]

[Editors’ note: the authors resubmitted a revised version of the paper for consideration. What follows is the authors’ response to the first round of review.]

Reviewer #1:In this study, the authors aim to show that hamsters may be a useful model of SARS CoV-2 long-term sequelae or "long COVID." The authors present useful data on many aspects of acute SARS CoV-2 in hamsters that are not well demonstrated in the literature such as gene expression of major regulators of inflammation, lipidomics, metabolomics, and antigen-specific CD4^+^ T cell responses which are important features of SARS CoV-2 in this model with translational potential if hamsters are used to test therapeutics aimed at preventing long COVID sequelae. However, the pathologic data that is presented to link both the lipidomics and metabolomics to extrapulmonary pathology suggestive of long COVID is weak. This study would greatly benefit from the addition of a treatment group or extended study timeline to support the claim that hamsters model key aspects of long COVID as seen in people. The lack of convincing pathology undermines the authors' model that lipidomic and metabolic changes are supportive of hamsters as a translational model of long COVID.

We are grateful to the in-depth review of our manuscript and critical comments to improve the manuscript. In line with the reviewers suggested, we have now included new experimental data from remdesivir therapeutic group (Figure S3D and S3E). Remdesivir is a previously known anti-viral drug that shows good efficacy in hamster. In line, we observed significant decrease in heart viral load in remdesivir treated challenged hamsters as compared to no remdesivir group. The serum cardiac troponin I level, which is a biomarker for CVC also was downregulated in remdesivir group. In addition, we have also included the ventricular mass/ body mass ratio as per the suggestion of the reviewers (Figure S3B). The assessment score for the histopathologies have been included in Figure 3. Wherever possible, we have now included blinded scores by trained pathologist for quantitation of the histological images.

Overall-there is lack of description on scoring systems for histology. The methods state that pathology assessment was performed by a histologist (?)--pathology should be reviewed by a blinded veterinary pathologist. The data presented to show cardiac hypertrophy, fibrosis, and inflammation is not convincing. The pathology images are very low resolution and the fibrosis and inflammation cannot be observed clearly (if present). Control (sham infected, procedure-matched) and age matched tissue specimens are not shown. Organ weights for spleen and heart are not shown (preferably need to see organ/body weight measurements). Heart weight is the gold standard for cardiac hypertrophy-without this measure cardiac hypertrophy cannot be diagnosed; at a minimum measurement of the interventricular septum should be shown-the images do not convincingly show differences over time.

We are thankful for the critical comments. We have now added a comprehensive statement on histological scoring system in the methodology section (Page 30 line number 573 to 575). Briefly, the blinded assessment and scoring (on the scale of 0 (no score) to 5 (highest score)) of each sample sections were performed by a trained pathologist.

Appropriate experimental control for Toulidine blue (Figure S2A) and MT staining (Figure S3C) has been added in the supplementary section along with histological scores for each pathological parameters has now been included as bar graph in the main figure panel (Figure 3D).

We have included the results from our new experiments showing spleen weight/ body weight (Figure 2B) and ventricular weight/body weight data (Figure S3B) in the manuscript. Moreover, we have also evaluated viral load in the heart and cardiac troponin-I serum levels which is a biomarker for cardiovascular diseases (Figure 3E and 3G). The new data has been described in the result section and Discussion section of the manuscript to support our findings.

Ln 155: The images of mast cells are not clear-how was this scored (counts/high power field?). Please show control animals (not just time 0).Likewise increased mucin staining in the GI tract is also not convincing. Please show controls over time.

Thank you for the comment. We have included an experimental healthy control Toluidine blue staining image in the supplementary figures (Figure S2A). Moreover, the statement defining the scoring for mast cells has been included in the figure legend and further described in the methodology section.

The GI tract mucin data has been removed from the manuscript since we found that the data was out of place and did not contribute to the main theme of the manuscript.

Reviewer #2:[…]Overall, the authors succeeded in showing how the hamster model can be useful to study systemic extra -pulmonic pathologies and systemic metabolic changes associated with SARS-CoV-2 infection. The data shown do not make a compelling case for the hamster model to study long term effects of infection outside the lung, given that the longest time point examined was d14, and the cardiac/vascular data were not robust. Furthermore, most of the cardiovascular complications that have been reported in COVD patients concern thrombosis, yet this report does not show any evidence of thrombotic complications in the hamsters.

We are grateful for the in-depth review of the manuscript and critical comments for improving the quality of the manuscript.

The possibility that cardiac cells in experimental animals are infected by SARS-CoV-2 should be tested. Recent reports that cardiac myocytes can be infected with SARS-CoV-2, including hamster myocytes (Chen, S et al. SARS-CoV-2 Infected Cardiomyocytes Recruit Monocytes by Secreting CCL2. Res Sq. 2020 doi: 10.21203/rs.3.rs-94634/v1; Marchiano, S et al. SARS-CoV-2 Infects Human Pluripotent Stem Cell-Derived Cardiomyocytes, Impairing Electrical and Mechanical Function. Stem Cell Reports. 2021 Mar 9;16(3):478-492. Doi: 10.1016/j.stemcr.2021.02.008.)

Thank you for indicating an important gap in our study regarding viral load quantitation in the heart samples. We have now included a heart viral load data in the main figure in support of our cardiovascular findings. Our heart viral load data suggests presence of virus in the cardiomyocytes of infected hamsters on day 2 and 4 post infection (Figure 3G).

Cardiac hypertrophy is best assessed by heart weight, and a second best way is to do careful quantitative analysis of left ventricular wall thickness on cross sections from all animals in each group. The photographs of two hearts.

Thank you for suggesting us to evaluate the heart weight/ body weight ratio as an indicator for ventricular hypertrophy. We have now included data from new experimental sets indicating changes in heart weight to body weight ratio in the supplementary figure (Figure S3B). We have also evaluated serum cardiac troponin-I levels in uninfected, infected and remdesivir group. Our results include elevated levels of cTnI levels in the serum of infected hamsters as compared to uninfected hamsters (Figure 3E).

Myocardial histopathology must be assessed and reported in quantitative fashion.

Thank you for suggesting us to evaluate the histogical data. We have now included a blinded score in the main figure panel for our histological images including the myocardial histopathology (Figure 3D). The methodology for scoring has been included in the methodology section.

In Figure S4 the images labelled Kidney are Liver, and vice versa. The kidney images are of poor resolution, and no glomeruli can made out in the d2 section. Sections from d7 and 14 should also be shown.

Thank you for pointing out this error in labelling. The other extra-pulmonary organ data including the GI tract histological data has been removed from the manuscript as we feel that this did not fit with the cardiovascular theme of the manuscript.

If there are activating anti-hamster CD3 reagents available to activate T cells, that would be a better (more selective, more physiologic) way for activating spleen T cells than PMA/odelling.

Thank you for suggesting to use T cell specific mitogen for activating splenocytes. Indeed, it would have shed more light on T cell clonal activation than using a heterogeneous activation by PMA/ Ionomycin. However, we did not have anti-hamster CD3 especially due to limitation in antibody resources for hamster study. We acknowledge this limitation in our study.

Reviewer #3:In this work the authors examine changes in the lungs and extrapulmonary organs in the setting of SARS-CoV-2 infection using the golden Syrian hamster model, with emphasis on cardiac changes. This is a well-established model to study SARS-CoV-2 infection. The authors also examine changes in splenic inflammatory related molecules, serum lipid molecules and serum metabolites.The major strength of the work is the attempt to perform detailed analyses of the extrapulmonary organs combined with splenic inflammatory changes, metabolomics and lipidomics in the setting of acute SARS-CoV-2 infection in an animal model.The major weakness of this work is that often the histologic changes being reported, particularly in the heart, are not well supported by the data provided. In addition, more rigorous statistical approaches could be employed, and there is no clear mechanism proposed to link the changes in the splenic inflammation, serum lipidomics, and serum metabolomics with the proposed cardiac changes.

We are grateful to the reviewer for the critically reviewing our manuscript and helping us improve the quality of the manuscript. We have now incorporated histological assessment data in the manuscript to support histology readout. We have also performed new experiments and evaluated cardiovascular markers and heart viral load to strengthen our findings.

1. There are numerous studies now on SARS-CoV-2 in the hamster. There should be much better effort relating this work to the published literature, particularly stressing what is novel and what is not. It is not clear if any of the pulmonary studies being reported here are actually novel.

Thank you for asking this question. It is true that SARS-CoV2 associated lung pathologies have been earlier reported in human and hamsters. However, in the present study we have tried to evaluate the other factors which are associated with lung injury such as Eotaxin, AGAR which have been previously shown to contribute to lung pathologies (Figure 1G). We have also evaluated the expression of mast cell functionality and it was found to be upregulated at the peak of infection. To our best of knowledge, these parameters have not been reported before for the hamsters SARS-CoV2 challenge study.

2. The authors claim: “Strikingly, SARS-CoV2 infection in the hamster leads to significant interstitial coronary fibrosis on 7 and 14 dpi characterized by thickening of ventricular walls interventricular septum.” And later: “The ventricular walls thickening at 7 and 14 dpi with SARS-CoV2 infection in GSH was marked by increased inflammation surrounding the coronary artery and elevated interstitial coronary fibrosis (Figures 4B, 4C).” However, the cardiac changes being reported are simply not supported by the images provided. Heart weights and wall thicknesses need to be measured and compared across the groups using statistics. Also, cardiac fibrosis and inflammatory cell infiltration should be quantified and comparted across the groups using statistics. Better images showing clear changes correlating with the quantitative analyses would be required.

We are grateful to the reviewer for asking us to evaluate the heart weight/ body weight ratio which is a standard method for evaluating ventricular hypertrophy. We have now included new sets of data from recently completed experiment and have incorporated new data on heart viral load, heart weight/body weight ratio and serum cardiac troponin-I levels. Furthermore, we have included the histological assessment data in the form of blinded scoring by trained pathologist. The method of scoring has also been described in the methodology section.

3. The authors state “…we reasoned that cardio-vascular pathologies of SARS-CoV2 in GSH may be related to changes in circulating lipid molecules (Bruzzone et 240 al., 2020; D. Wu et al., 2020). Indeed, serum lipid profile showed elevated cholesterol, TGs,HDL, LDL and VLDL levels at 4 dpi (Figure 4D).” However, the changes in serum lipids were very transient and unlikely to rapidly cause cardiac fibrosis. There is no clear logical mechanism being present linking the serum lipid levels with the proposed cardiac changes.

We are grateful for the valuable suggestion. We have now incorporated new data set from recent studies in which we have evaluated both viral load and serum cardiac troponin-I level in circulation that could be an valuable markers for validating cardiovascular pathologies in SARS-CoV2 infected hamsters. Furthermore, we have also added a histological assessment data based on blinded random scoring by trained pathologist to support the quantitative assessment of cardio-vascular histopathologies.

4. For all of the studies performed including the lipidomics and metabolomic studies the authors simply us a P-value of less than 0.05 as being statistically significant with no adjustment for multiple assessments or consideration of FDR. A more sophisticated statical approach is needed.

We appreciate the concern raised by the reviewers. The FDR correction has been incorporated for the omnics data and a statement describing FDR correction has been incorporated in the methodology section (Page 34, line number 658).

5. The spleens need to be weighed and the weights compared across the groups using statistics.

Thank you for this critical suggestion. We have not included the spleen weight/ body weight data in the main figure panel to support the splenomegaly condition arising due to SARS-CoV2 infection (Figure 2B).

6. There are frequent statements in the discussion that simply do not match the data shown. For example, the authors state: “Our data show that lung pathologies long persists after the virus clearance from the lung since we observed a high pathological score on 7 dpi.” However, the authors own data show that the virus is still present at 7 dpi.

We regret for the ambiguity in the statement. We have now corrected the statement to make it simpler and unambiguous.

7. All figure panels need proper labels, particularly both axes of any plots.

Thank you for raising this concern. We have rechecked the figures of the manuscript and have added the axis wherever it was missing.

8. In Figure S4, the degree of mucin staining in the images does not appear to match that shown in the bar graphs. The scale for the scale bars is not legible. For Panel B the images marked “liver” are not liver, and the images marked “kidney” are not kidney.

Thank you for pointing out the limitations of the study. Since we have now revised the manuscript focusing on the cardiovascular and lipido-metabolomics changes we feel that other extra-pulmonary pathologies do not fit with the main theme of the study. Therefore, we have removed histology data for the other organs from the manuscript.

9. There should be some rational mechanism linking the splenic inflammatory cell changes, lipidomics and metabolomics with the proposed cardiac changes.

Thank you for your comment. We have improved the manuscript in line with the suggestions made and have tried to connect the immunological and cardiovascular changes with other parameters studied. In line with this, we have improved the introduction and Discussion section focusing on the pathological changes associated with SARS-CoV2 infection and molecular factors that maybe involved in the pathologies. For example, inflammatory cytokines contribute to lung pathologies and further IL6 and some other cytokines and chemokines have also been described as contributors for CVC. CVC is usually associated with injuries to cardiomyocytes which may happen due to accumulation of cholesterol, LDLs and LCFAs in the circulation. Furthermore, N-acetylneuraminate metabolite levels have been previously linked with CVC condition in clinical settings.

[Editors’ note: what follows is the authors’ response to the second round of review.]

The manuscript has been improved but there are some remaining issues that need to be addressed, as outlined below:1) Revise statistical analyses using non-parametric approaches as suggested.2) Minimize or remove connections made to long COVID- the evidence is not strong enough that this is a model.3) Include recent references as suggested.4) Extensive editing of grammar and syntax is absolutely necessary.5) Make MS readouts legible.

Thank you for your valuable comments which will help in improving the quality of the manuscript. We have now made necessary revisions throughout the manuscript in line with the reviewers comments. Briefly, the statistical analysis of the data now shows non-parameteric analysis and the exact p values of the data has been shown in the figure legends. We have revised or removed long-COVID term wherever applicable throughout the manuscript. All the recent references suggested by the reviewers has been cited in the manuscript. Moreover, we have thoroughly worked on the grammar and syntax of the manuscript and have tried to improve it to our best capabilities.

Reviewer #1:This manuscript nicely highlights SARS CoV-2 effects on the heart in the Syrian hamster model, particularly from the virologic, transcriptional, and metabolic standpoint. The data clearly highlights metabolic changes that support the exploration of using the hamster model for cardiovascular complications of COVID. The data for histopathologic evidence for cardiovascular changes remain unconvincing at the timepoints shown. I believe a longer-term study following hamsters for greater than 14 days would be more biologically relevant for histopathologic changes. Regardless, the biochemical and immunologic data support the use of this model for studying cardiovascular changes.

Thank you very much for providing in-depth review and helping in improving the quality of the manuscript. It would have been indeed interesting to do a longitudinal study for SARS-CoV-2 infection in hamster looking at cardiovascular changes beyond 14 days window. We accept this as a limitation to our study. Nonetheless, we believe that the findings of the manuscript would be crucial for constructing further studies related to cardio-vascular pathologies in hamsters.

The authors have addressed all the concerns raised. Unfortunately, the histopathologic data, although shown as significant via scoring, is still not biologically convincing. The heart weight to body weight index tracks for infected animals only. The ventricular width should have been done quantitatively and not semi quantitatively (scoring) to be convincing. The appropriate statistical comparison would be heart mass/body mass of infected to uninfected animals at each time point assayed. Also, the semiquantitative scoring of inflammation and fibrosis does not match up with what can be seen by eye. Lastly, the scoring metric for pathology should be clearly outlined in the figure legend (i.e. 1-5 minimal, mild, marked—for what features). What is the total score possible?Regardless—the cytokine, lipid, and metabolic profiling and viral load data from the heart is very interesting and supports the further exploration of this model for studying cardiovascular effects of COVID.There is still substantial copy editing needed for this manuscript. Many sentences do not make sense and grammar needs substantial improvement prior to publication.

Thank you for your valuable suggestions. We have carefully revised the manuscript wherever possible in line with the suggestions made. We have thoroughly worked on the grammar and have improved it to the best of our capabilities. At this point of time, we do not have quantitative evaluation of ventricular wall thickening. However, our ventricular mass/ body mass data suggests that the day 7 and day 14 ventricular mass/ body mass ratio was significantly higher than that of the healthy control indicating ventricular hypertrophy (Figure 3—figure supplement 1B). We have also improved the methodology section on the semi-quantitative scoring method of the histological sections (Histology section of methodology). The histological score for respective pathologies were given on the scale of 0-5, where score 0 indicated the absence of pathological signs, while score 5 is indicative the highest pathological changes at given time points. This information now been included in the figure legends 1 and 3 as per the suggestions.

Reviewer #2:The authors have studied SARS-CoV-2 infection in golden hamsters. Evidence is provided for viral RNA in the heart and for many alterations in the circulation, discovered by metabolomics and lipidomics, including many that have been linked to cardiovascular disease in humans. The novel aspect of the studies presented are the studies utilizing lipidomics and metabolomics and the abrogation of heart disease after Remdesevir. This is a revised manuscript and most of the concerns of the previous reviewers have been largely addressed. However, there is at least one other relatively recent study that the authors have failed to make note of, (and that was published in July 2021), that has also observed myocardial inflammation and evidence for viral RNA in the heart. So, some of the findings have been independently confirmed and that is viewed as a strength rather than as a limitation. The other study made simple, direct links between the presence of viral RNA in a tissue and the events that lead to cardiac inflammation.The authors have made some important improvements to the first submission especially in terms of better documenting the cardiac involvement in the model.At this point in time changes like the increase in circulating sialic acids are interesting but most of the metabolic and lipidomic changes are best attributed to systemic inflammation as well as to heart damage. Given the time frame, it is difficult to consider these alterations as causal for the heart inflammation seen but these changes could be of relevance when one considers the cardiovascular complications of clinical COVID-19 that fall into the PASC spectrum.

Thank you very much for providing in-depth review and helping in improving the quality of the manuscript.

The authors should more comprehensively reference earlier papers on the use of hamsters as a model for SARS-CoV-2 including studies in 2020 from Rosenke et al. (PMID 33251966) and Boudewijns et al. (PMID 33203860). However, they should, at least during this submission, have quoted a manuscript showing cardiovascular disease in a similar model that was published in July 2021. This study published about four months ago from Francis and colleagues. (PMID 34265022) should certainly have been discussed because it in part supports the authors’ studies and demonstrated inflammation in the heart. In the studies of Francis et al., there was viral RNA in the heart though live virus was not recovered from this organ, and there was evidence provided for Type I interferon dysregulation, eosinophilic myocarditis, inflammatory cytokines, T cell infiltration and activation, though the evidence suggested the absence of TH2 cells and T regs.

Thank you very much for providing in-depth review and helping in improving the quality of the manuscript. We have tried our best to address the concerns in the manuscript and have updated the references list as suggested.

There remain issues related to the authors’ lack of care with syntax and grammar that still need to be taken care of and careful proofreading of the manuscript should be undertaken.

We are grateful for the insightful review of our study and helping in improving the quality of the manuscript. We have thoroughly worked on the grammar of the manuscript and have addressed the points raised along with other grammatical corrections to the best of our capabilities.

Reviewer #3:– A central issue is that depending on the dose, SARS-CoV-2 is cleared in 5-6 days in the hamster model. Authors need to defend and define how can this is appropriate for odelling late phase effects of COVID-19 in humans.

Thank you for raising this concern. We agree with the comment that further longitudinal studies would be required for better correlating the cardio-vascular pathology in hamster with the long-COVID symptoms in human. We have therefore, decided to remove the use of long-COVID usage from the manuscript to make it more coherent for the audience.

– Throughout their figures the authors indicate that one-way ANOVA is used to compare the groups but do not specify the post-hoc test being used to compare the individual groups. E.g. Dunnett’s test would be the most appropriate for comparing each group to the control (e.g. time zero). However, ANOVA with post-hoc should be applied only to parametric data (i.e. normally distributed measurement data). However, the scoring data will be non-parametric and will need to be compared using a non-parametric test.

Thank you for your comments and helping us improve the quality of the manuscript. We have revised the statement on statistics throughout the manuscript with disclosure on parameteric/ non-parametric analysis in the figure legends of the main panel as well as supplementary panel. Furthermore, we have mentioned the statistical test used for the analysis as suggested.

– In their results (line 158), the authors equate recovery with elevated antibodies at day 14. Elevated is antibodies is expected (e.g. would get same result in vaccination) and should not be taken as a proxy for recovery.

Thank you for raising this important point. Indeed, neutralizing antibody response is better correlated with the protective immunity against SARS-CoV-2 infection and poorly correlated with the recovery response against COVID-19. In light of the suggestions, we have accordingly revised the manuscript and removed the ambiguous statement.

– Lines 187-189. authors contend that a cytokine storm is in operation during acute infection. Circulating levels of IL-6 is one of the key cytokines that predicts storms in human COVID-19. Why was this not measured? IL-6 mRNA was measured later, but the authors should measure the actual level of IL-6 in circulation.

Thank you for pointing this out. We have now included additional IL-6 ELISA data in the main figure (Figure 3F) and supplementary figure (Figure 3—figure supplement 1E) to support our findings.

– It seems that in the biochemical analysis vs LC-MS/MS analysis paint different pictures regarding the time course of lipid biosynthesis. It seems that in the latter case, the authors are more interested in reporting changes, rather than the trajectory of those changes (e.g. in Figure 4G and F it's not possible to tell what is up and what is down). This paints a confusing picture. It also doesn't help that the text on Figure 4F is too small to read.– Use Kruskal-Wallis test for comparing non-parametric score/count data.

We are grateful to the reviewer for making valuable suggestions, we made the changes in revised manuscript as suggested. We agree with the reviewers that the labels for figure 4F is too small to read, this is due to higher numbers of lipid molecules showing significant downregulation. To accommodate reviewer’s suggestions, we have re-created the figures with updated colour schemes for better clarity in the changes in lipid profile observed. We have also provided a higher resolution heatmap (full figure) as separate supplementary material for better clarity and readability.

– Would suggest disentangling the proposed parallels between late stage effects seen him COVID-19 in humans with SARS-CoV-2 infection in hamsters, unless longer term effects well outside the two week sampling window can be demonstrated.

We thank the reviewer for an insightful comment. We accept that longitudinal follow up of the hamsters beyond 14 days post infection would have added value in drawing parallel of late stage effects seen in COVID19 in humans especially in the context of cardio-vascular pathologies. We have therefore, decided to remove the term “long-COVID” usage from the manuscript to make it more coherent for the audience.